# Longitudinal liquid biopsy identifies an early predictive biomarker of immune checkpoint blockade response in head and neck squamous cell carcinoma

Binbin Wang [1,14], Robert Saddawi-Konefka [2,3,4,5,14], Lauren M. Clubb[3], Shiqi Tang[6], Di Wu[7], Sumit Mukherjee [1], Sahil Sahni [1], Saugato Rahman Dhruba [1], Xinping Yang[8], Sumeet Patiyal[1], Chi-Ping Day [1], Parth A. Desai[9], Clint Allen [8], Kun Wang [10,11,12] ✉, J. Silvio Gutkind[3,4,13] ✉ & Eytan Ruppin [1] ✉

Immune checkpoint blockade (ICB) has improved outcomes for patients with head and neck squamous cell carcinoma (HNSCC), but predictive biomarkers remain limited. Here, we use a time-resolved, multi-omic approach in a murine HNSCC model to characterize peripheral immune responses to ICB. Single-cell transcriptomics and T/B cell receptor analyses reveal early on-treatment expansion of effector memory T and B cell repertoires in responders, preceding tumor regression. These dynamic immune features inform a composite transcriptional signature that accurately predicts ICB response in independent human HNSCC cohorts. LiBIO outperforms existing biomarkers and generalizes to melanoma, non-small cell lung cancer, and breast cancer without retraining. These findings suggest that early treatment-induced changes in circulating immune repertoires reflect the host's capacity to mount an effective antitumor response. This work provides a framework for leveraging transient peripheral immune dynamics to develop non-invasive, high-fidelity biomarkers for response to immunotherapy across cancer types.

Immune checkpoint blockade (ICB) has shown well-established efficacy in the treatment of numerous cancer types, including head and neck squamous cell carcinoma (HNSCC). However, only a subset of patients experiences a beneficial response, and these rates can vary widely among different cancer types[1]. This variability highlights the urgent need to identify reliable biomarkers that can predict which patients are likely to benefit from ICB treatment.

The FDA has approved several key biomarkers associated with response to ICB, including tumor mutation burden (TMB), PD-1 gene expression, and microsatellite instability (MSI)[2]. Higher TMB is associated with an increased likelihood of response, likely due to the greater availability of neoantigens that can be targeted by the immune

system[3,4]. PD-1 protein expression is one of the commonly established biomarkers of ICB response across multiple cancer types, including non-small cell lung cancer (NSCLC) and melanoma[5]. MSI is another widely used biomarker of ICB response. Tumors classified as MSI-high tend to respond favorably to immunotherapy[6]. In HNSCC, two FDA-approved immunotherapy biomarkers are utilized: tumor mutational burden (TMB) and the programmed death-ligand 1 (PD-L1) combined positive score (CPS). The CPS measures the expression of PD-L1 on both tumor cells and immune cells, thereby providing a comprehensive assessment of the tumor's interaction with the immune system[7–10]. Beyond these FDA-approved biomarkers, numerous other genomic and transcriptomic ICB biomarkers have been identified for different

cancer types[11,12]. For instance, TIDE[13] identified a T cell exhaustion signature that is predictive of ICB response in melanoma. IMPRES[14] characterized gene pairs in the immune checkpoint pathway and identified pairs associated with ICB response. Greater numbers of tumor-infiltrating lymphocytes (TILs), inferring a pre-existing immune response within the tumor microenvironment, has been shown to correlate with better immunotherapy outcomes[15–17]. Additionally, gene expression profiles reflecting immune activation and cytokine levels can provide further insights into the likelihood of response[18–20]. Despite this progress, the predictive accuracy of existing biomarkers remains limited[11]. Additionally, these signatures are designed to characterize tumor-intrinsic and microenvironmental features, necessitating the collection of tumor tissue samples through biopsies or surgical procedures.

In contrast to invasive tumor sampling that introduce both cost and risk of morbidity, liquid biopsies offer a minimally-invasive alternative that can be performed with relative ease. Liquid biopsies are designed to capture two fundamental types of data: (1) information on the cancer cells themselves, including circulating tumor cells, circulating tumor DNA, exosomes, and more[21–23]; (2) information on immune cells circulating in the blood, whose abundance has been reported to be associated with ICB response, including monocytes, dendritic cell, myeloid cell, lymphoid cell[21,23], and the neutrophil-to-lymphocyte ratio[24]. Most of these abundance signatures were developed as pre-treatment biomarkers and focused mainly on NSCLC and melanoma[24,25]. The antitumor immune response is inherently unidirectional and integrates the peripheral immune system through a sequential pathway: tumor → regional lymphatics → periphery → tumor. This framework underscores the need to refocus on a systematic investigation of peripheral immune events at more advanced time points during the treatment window with biomarker identification serving as the cornerstone of this effort.

Here, we conduct both bulk and single-cell transcriptomic sequencing and single-cell TCR sequencing at four time points across the ICB treatment span in a mouse HNSCC model. These include one pre-treatment time point and three on-treatment time points. We utilized these data to identify effective liquid biomarkers for anti-PD-1 treatment in human HNSCC and specifically learn which time point is the most informative and predictive with obvious preference to earlier time points. By analyzing the mouse data, we charted the clonal expansion of both T and B cells during ICB treatment, observing that responders and non-responders show a significant difference in the dynamics of their clonal repertoire expansion and pruning. Markedly, the strongest response to the ICB was detected in the earliest on-treatment time point, showing significant transcriptomic and TCR clonality differences between responders and non-responders. This aligns with the timing of tumor shrinkage initiation. Beyond our preclinical findings, we proceed to show that the ICB response related signatures derived using mouse data can successfully predict ICB response in HNSCC patients, demonstrating their potential clinical application. To our best knowledge, this is the first study that systemically characterizes the dynamic changes of circulating immune cells in the blood during ICB treatment across four sequential time points, associating these alterations with ICB response in patients.

## Results
### Study overview
We employed our previously characterized 4MOSC carcinogen-induced orthotopic HNSCC preclinical model[26,27] to evaluate immune dynamics during ICB therapy. Two cohorts of mice were used, with one cohort comprising 45 mice for bulk RNA sequencing (RNA-seq) and the other comprising 16 mice for single-cell RNA sequencing (scRNA-seq) and single-cell TCR sequencing. Lesions developed consistently at expected time points[26,27] (Fig. 1A, B). Blood samples were collected at four key intervals: pre-treatment (Day 4) and three on-treatment time

points (Days 9, 17, and 24), aligned with the administration of anti-PD-1 therapy. Tumor growth and immune responses were monitored longitudinally to assess treatment efficacy (Fig. 1C, D). Bulk RNA-seq provided an overview of immune dynamics in the larger cohort, while scRNA-seq and scTCR sequencing in the smaller cohort delivered high-resolution insights into immune repertoire changes, supporting the study's focus on peripheral immune biomarkers for predicting ICB outcomes.

The 4MOSC model used in this study is a well-characterized, carcinogen-induced, orthotopic HNSCC model that exhibits a reproducible mixed-response to anti-PD-1 therapy (~30–40%), consistent with clinical response rates observed in human HNSCC. This model was originally described in Wang et al., Nat Comm 2019[27] and further refined in Saddawi-Konefka et al., Nat Comm 2022[26], where we demonstrated that response heterogeneity is driven by differences in immune activation and lymphatic function, rather than tumor-intrinsic clonal variation.

### Single-cell analysis uncovers specific temporal changes in blood T and B cell abundances differentiating ICB responders from non-responders
To profile the immune landscape and its dynamic changes during ICB treatment, we performed unsupervised clustering of cells from 63 samples collected from 16 mice across the 4 time points described above. This analysis identified 9 major cell types (SF1A, B). Among mononuclear cells, CD8+ T cells, CD4+ T cells, NK cells, and B cells are the dominant cell types in the blood samples. Comparisons of on-treatment to pre-treatment time points revealed significant increases in the abundance of CD8+ T cells, CD4+ T cells, and B cells following ICB treatment (one-tailed Wilcoxon rank-sum test, $p < 0.05$) (Fig. 2A, B). In contrast, the relative abundance of neutrophils decreased over time (Fig. 2B).

To identify T cell subpopulations associated with ICB response, we further identified subsets of CD8+ T cells and CD4+ T cells by unsupervised clustering. To ensure accurate clustering and annotation of these subpopulations, we projected the CD8+ and CD4+ T cells onto a reference mouse T cell database (SF 1C, D). This analysis identified two dominant subtypes of CD8+ T cells including naïve CD8+ T cells and effector memory CD8+ T cells ($T_{em}$) (SF 1E, F), and three major subtypes of CD4+ T cells including naïve CD4+ T cells, Type 1T helper (Th1) cells and regulatory T cells (Tregs) (SF 1G, H). Among these T cell subpopulations, $T_{em}$ (Fig. 2C) and Th1 CD4+ T cells increased following ICB treatment in responders (SF 2A). A monotonic increase in these two cell types was observed across all three on-treatment time points in responders, resulting in the highest accumulation levels at the late on-treatment time point (Day 24) (Mann–Kendall test, P-value for $T_{em}$ $5.6 \times 10^{-4}$, P-value for Th1: $9.0 \times 10^{-5}$,) (Fig. 2C, D and SF 2A, B). This monotonic increase is not observed in non-responders following ICB treatment: While there was a modest increase in $T_{em}$ at the early on-treatment time point (Day 9), this was followed by a sharp decline at the middle time point (Day 17) (Fig. 2C).

B cell abundance increased in both responders and non-responders following ICB treatment. There was no significant difference between responders and non-responders at the pre-treatment (Day 4) and late on-treatment time points (time points 4) (Fig. 2E). However, we observed a modest but statistically significant earlier increase in B cells among responders at the early on-treatment time point (Day 9) (one-tailed Wilcoxon rank-sum test, $P = 0.027$). In contrast, non-responders exhibited a delayed B cell accumulation that became apparent only at the middle on-treatment time point (Fig. 2F). These findings align with a previous study[28] that demonstrated a predictive role for B cells in response to ICB in HNSCC.

Overall, distinct dynamic changes in immune cell composition were observed in blood samples following ICB treatment, highlighting key differences between responders and non-responders. Responders

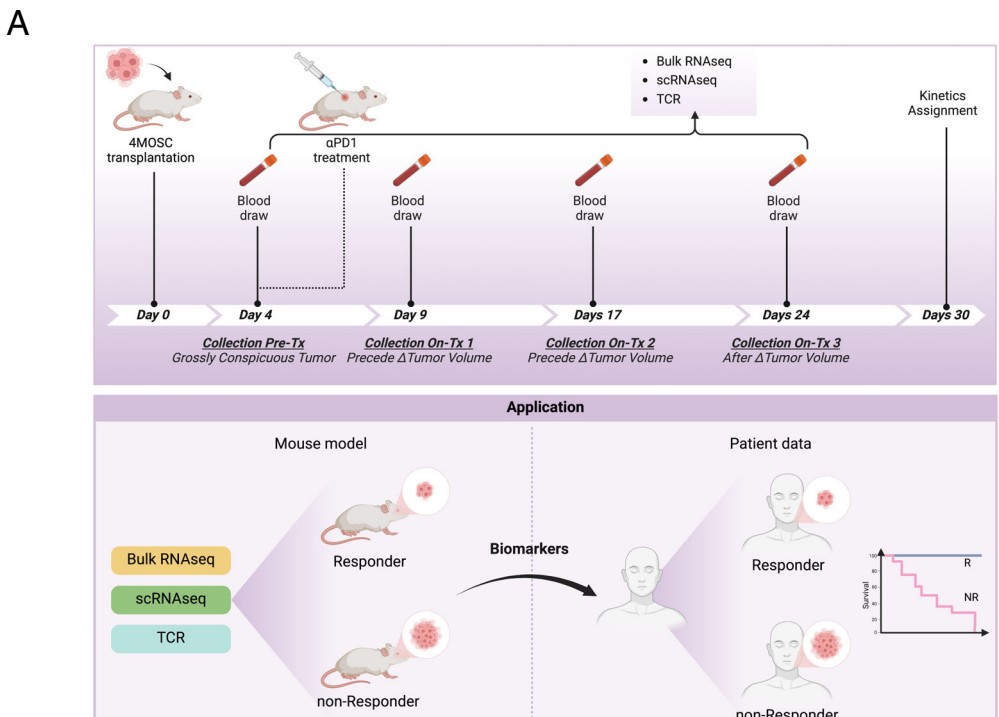

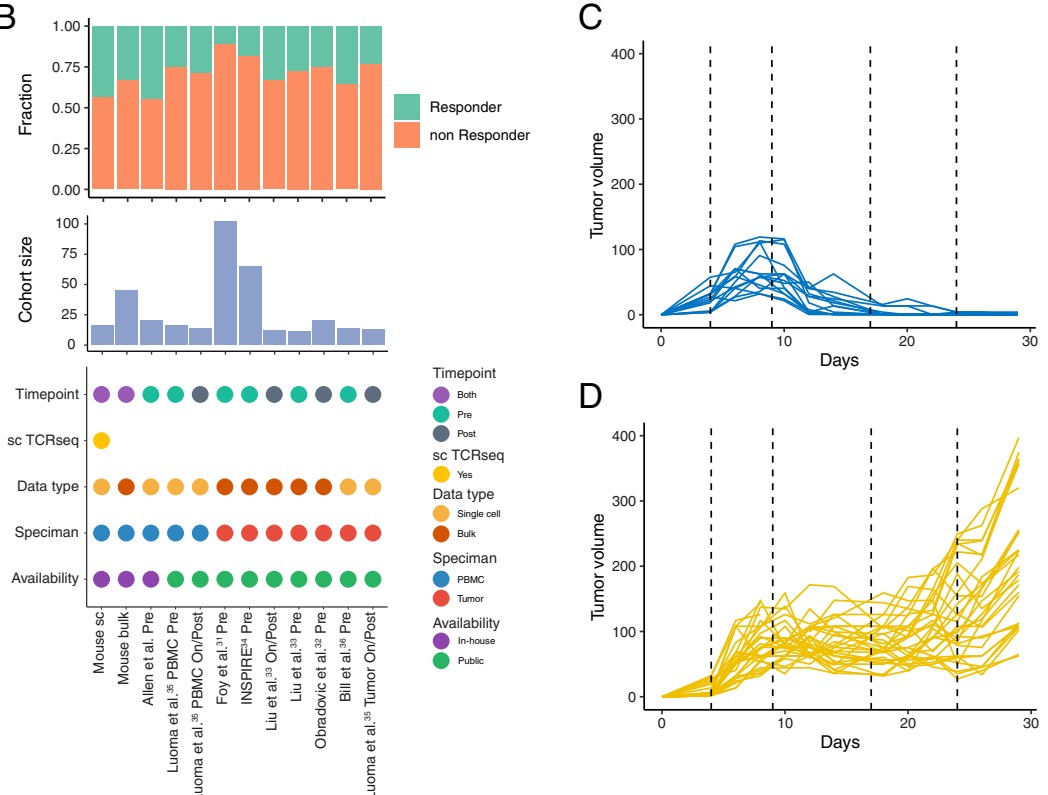

demonstrated an accumulation of T cell-mediated cytotoxic immune cells and a downregulation of suppressive immune cells contributing to enhanced immune activation. Conversely, non-responders initially accumulated immune response-related cells at the early on-treatment time point, followed by their decrease. Moreover, B cell accumulation occurred at different time points between responders and non-responders, further underscoring the distinct immune dynamics

associated with ICB response. The accumulation of $T_{em}$ and B cells began early in the on-treatment phase (Day 9) and negatively correlated with tumor shrinkage, which also started at Day 9 in responders (Fig. 1C, D). These results not only highlight the crucial role of $T_{em}$ and B cells in ICB response but also illustrate that the early on-treatment phase serves as a watershed between responders and non-responders in HNSCC.

**Fig. 1 | Overview of experimental design and data. A** Schematic of the experimental design. Blood was collected at four time points: one pre-treatment time point (four days after cancer cell transplantation) and three on-treatment time points (days 9, 17, and 24). Anti-PD-1 treatment was initiated on day 4 following the first blood collection. The response to ICB treatment for each mouse was assessed based on RECIST criteria. This graphical abstract was created with BioRender.com. **B** Summary of in-house and publicly available HNSCC datasets used in this study. Three in-house datasets were generated, including an RNA-seq dataset from 45 mice across four time points, consisting of 15 ICB responders and 30 non-responders. Additionally, single-cell RNA-seq and TCR sequencing were performed

on an independent cohort of 16 mice, comprising 7 responders and 9 non-responders. Another in-house dataset includes a single-cell RNA-seq dataset from ICB-treated HNSCC patients, containing 20 patients (9 responders and 11 non-responders). Furthermore, two publicly available single-cell ICB-treated datasets and four publicly available bulk RNA-seq ICB datasets were incorporated into the analysis. **C** Dynamic changes in tumor volume across time points in responders. The dashed line indicates the four time points at which whole blood were collected. **D** Dynamic changes in tumor volume across time points in non-responders. The dashed line indicates the four time points at which PBMCs were collected. Source data are provided as a Source Data file.

## ICB treatment induces T cell and B cell clonal expansion that is predictive of ICB response

Single-cell TCR sequencing data enable the detection of expanding T cell clones. Analysis of TCR sequencing data uncovers a clonal expansion in CD8+ T cells within the blood $T_{em}$ subpopulation during ICB treatment (Fisher's exact test, odds ratio: 30.5, $p < 2.2 \times 10^{-16}$) (Fig. 3A, B). Moreover, these expanded clones were significantly enriched in responders compared to non-responders (Fisher's exact test, odds ratio: 1.73, *P*-value: $3.75 \times 10^{-5}$) (Fig. 3C) (SF 3A). The strongest expansion was observed at early on-treatment time points (one-tailed Wilcoxon rank-sum test, $p = 0.027$), followed by a dramatic decrease at the middle on-treatment time point (Fig. 3D). ICB treatment-induced clonal expansion is followed by subsequent pruning at the late on-treatment time point (Fig. 3D). In CD4+ T cells, the expanded clones are primarily enriched in the Th1 subpopulation (SF 3B, C), with the strongest expansion occurring at early on-treatment as well, without significant differences between responders and non-responders (SF 3D).

To validate these findings, TCR analysis was performed using bulk RNA-seq data from an independent mouse cohort. The bulk TCR analysis results confirmed that T cell clonal expansion begins at early on-treatment time points and reaches completion by the late on-treatment time point, with responders exhibiting significantly stronger expansion than non-responders (one-tailed Wilcoxon rank-sum test, $p = 0.026$) (Fig. 3E). BCR analysis using bulk RNA-seq data further demonstrated significant B cell clonal expansion following ICB treatment (Fig. 3F). Responders exhibit increased B cell expansion at early (one-tailed Wilcoxon rank-sum test, $p = 0.00024$) and middle on-treatment time points (one-tailed Wilcoxon rank-sum test, $p = 0.031$), before returning to pre-treatment levels at the late time point (Fig. 3F). Interestingly, both T and B cell clonal expansion at early on-treatment time points exhibit strong predictive power for ICB response (Fig. 3G). Specifically, B cell clonal expansion demonstrates superior predictive ability compared to T cell clonal expansion at early on-treatment time points (Fig. 3G). This aligns with the timing of tumor shrinkage initiation in the responders (Fig. 1C, D) as well as the changes in cell abundance estimated from the single-cell data (Fig. 2D, F). Notably, B cell clonal expansion is predictive not only at the pre-treatment time point but also at two on-treatment time points, further underscoring the critical role of B cells in mediating the ICB response in HNSCC (Fig. 3G) (SF 4A–D).

Taken together, these findings underscore three key points: first, ICB-induced T and B cell clonal expansion occurs predominantly at early on-treatment time points and is followed by subsequent pruning by the late on-treatment time point despite continued PD-1 treatment; second, responders exhibit significantly stronger expansion in both T and B cells compared to non-responders; and third, this clonal expansion can be reliably detected in the blood using either single-cell TCR sequencing or bulk RNA-seq.

## Serial Bulk RNA-seq analysis identifies the early on-treatment time point as most predictive of ICB response in the blood

To characterize the dynamic changes in biological functions following ICB treatment, we applied the fuzzy *c*-means algorithm to identify patterns of dynamic alterations in the expression of the different circulating immune cell types from bulk RNA-seq data. This analysis

identified eight characteristic clusters (C1–C8) of gene expression patterns in both responders and non-responders (Fig. 4A, B). In responders, genes within five clusters (C1,3,4,5,8) were upregulated at the Day 24 compared to pre-treatment, whereas in non-responders, six clusters (C1,2,4,5,6,7) were downregulated comparing these time points. The upregulated clusters in responders were related to B cell immunity, immunoglobulin production, and associated pathways, including protein-RNA complex assembly, mRNA processing, and mitochondrial gene expression (Fig. 4A, B). In contrast, genes related to B cell activation and immunoglobulin production were downregulated in non-responders. These findings suggest that B cells play a crucial role in the response to ICB treatment and could serve as predictive biomarkers for ICB response in HNSCC[28]. In addition to the divergent dynamic changes between responders and non-responders, immune response pathways were activated early in both groups. However, in responders, additional pathways, such as myeloid leukocyte activation and chemokine production, were also activated to enhance and support the immune response. In non-responders, pathways related to innate immune response, leukocyte-mediated cytotoxicity, and cell killing were downregulated, thereby suppressing the immune response (Fig. 4A, B). Notably, the highest number of differentially expressed genes between responders and non-responders was observed at the early on-treatment time point (Day 9), highlighting its potential functional and predictive significance (SF 5A).

To further assess the association between each time point and treatment outcome, we trained a lasso regression-based ICB predictor using bulk RNA-seq data from each time point. The five-fold cross validation reveals that the predictor trained at early on-treatment time point (Day 9) achieves the same performance as the late on-treatment time point (Day 24), indicating that the former can serve as an optimal time point for treatment efficacy (Fig. 4C). The early on-treatment time point (Day 9) emerges as the optimal window for assessing treatment efficacy, as it precedes observable tumor shrinkage and captures critical early immune dynamics associated with response. To investigate whether dynamic changes across time points could further enhance prediction accuracy, we built a predictor based on gene expression changes between time points. The changes between time points 4 and 2 outperformed any single-time point and dynamic change-based predictors in predicting ICB response, demonstrating that temporal dynamics may provide valuable insights for predicting ICB response but do not provide a marked increase in the prediction power over that of the early on-treatment time point (Fig. 4D).

Taken together, these results highlight distinct dynamic gene expression patterns between responders and non-responders, with differences in immune response and B cell activation. Furthermore, the observed temporal changes in gene expression serve as promising biomarkers for predicting ICB response, with the early on-treatment time point emerging as an optimal time for treatment assessment.

## Analysis of mouse SC data identifies predictive blood $T_{em}$ and B cell signatures that are predictive of ICB response in both bulk and SC expression cohorts of human patients

We demonstrated that ICB treatment induces clonal expansion and proliferation of $T_{em}$ and B cells, particularly at early on-treatment time

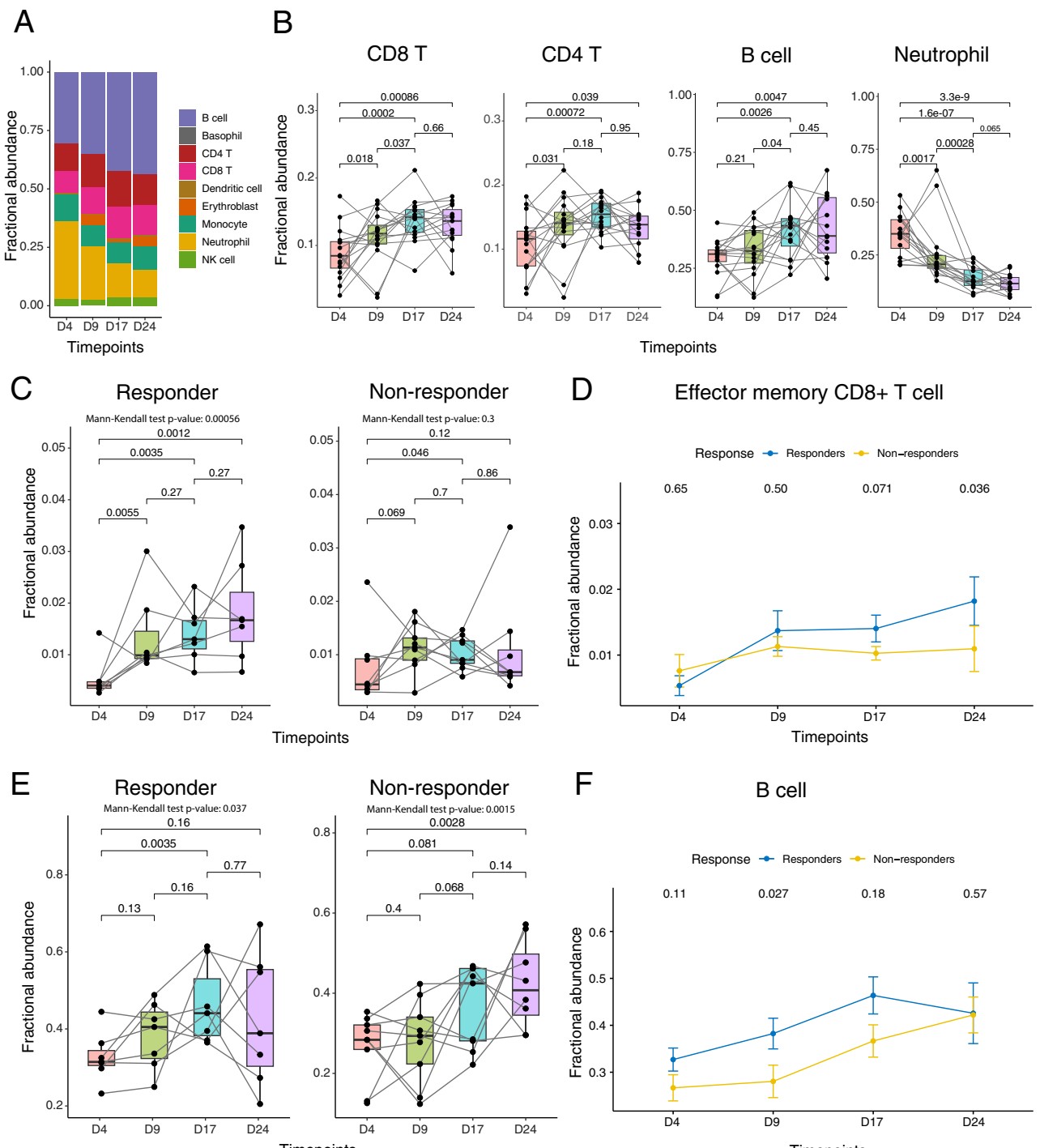

**Fig. 2 | Temporal changes in cell abundance in blood following ICB treatment.**
**A** Dynamic changes in the abundance of major cell types in blood. **B** Box plots showing the abundance changes of four cell types significantly influenced by ICB treatment ($n = 16$ biological replicates; each dot represents one mouse).
**C** Abundance changes of effector memory CD8+ T cells in responders ($n = 7$) and non-responders ($n = 9$) across four time points. Each dot represents an individual mouse. **D** Abundance differences of effector memory CD8+ T cells between responders and non-responders. **E** Abundance changes of B cells in responders ($n = 7$) and non-responders ($n = 9$) across four time points. Each dot represents an individual mouse. **F** Abundance differences of B cells between responders ($n = 7$)

and non-responders ($n = 9$). In box plots (**B**, **C**, and **E**), the center line indicates the median; the box spans the interquartile range (IQR, 25th to 75th percentile); whiskers extend to values within 1.5× IQR from the quartiles; and each dot represents one biological replicate (a single mouse). In (**D** and **F**), dots represent the mean, and error bars indicate the standard error of the mean (± SEM). The unit of study is the individual mouse. Statistical significance was assessed using a one-tailed Wilcoxon rank-sum test unless otherwise noted. The Mann–Kendall test was used to evaluate monotonic changes across time points. For all panels, the X-axis represents time points, and the Y-axis represents the fraction of cells out of the total measured. Source data are provided as a Source Data file.

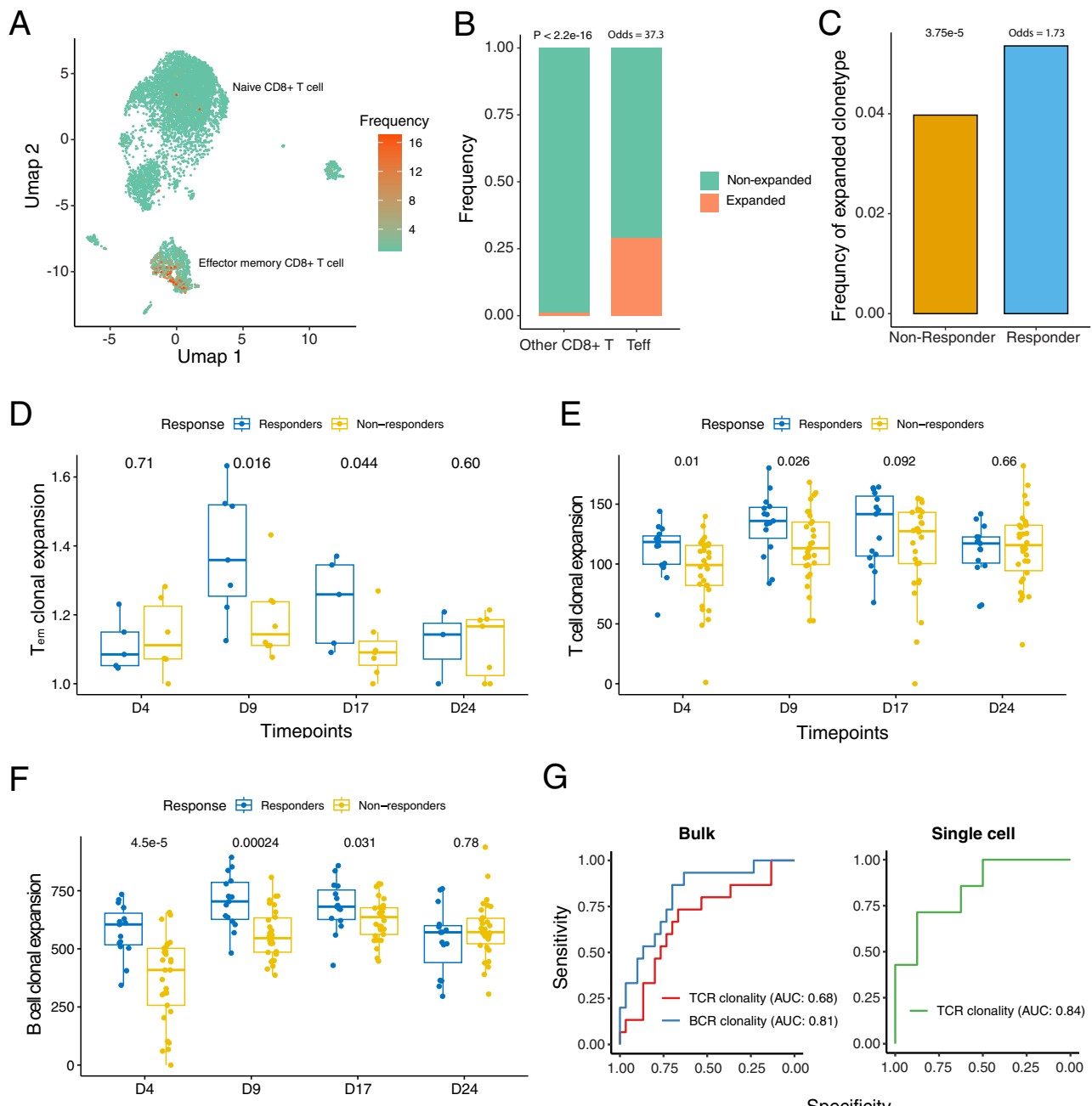

**Fig. 3 | ICB treatment induces T cell and B cell clonal expansion. A** UMAP of CD8+ T cells, with each dot representing a single-cell and colored by clone size. **B** Distribution of expanded clones (clone size ≥2, represented by red bars) and non-expanded clones (clone size = 1, represented by green bars) between effector memory CD8+ T cells and other CD8+ T cells. *P*-value and odds ratio were calculated using Fisher's exact test. **C** Distribution of CD8+ T cell clone sizes between responders and non-responders across four time points. Statistical significance was determined using Fisher's exact test. **D** Comparison of effector memory cell ($T_{em}$) clonal expansion estimated using single-cell TCR-seq data, between responders (*n* = 7) and non-responders (*n* = 9). The *y*-axis represents the average size of $T_{em}$ clones, where higher values indicate greater clonal expansion. Each dot represents an individual sample. Statistical significance was determined using a one-tailed Wilcoxon rank-sum test. **E** Comparison of T cell clonal expansion estimated using bulk RNA-seq data, between responders (*n* = 15) and non-responders (*n* = 30). The *y*-axis represents the scaled Simpson index, where higher values indicate greater clonal expansion. Each dot represents an individual mouse. Statistical significance was determined using a one-tailed Wilcoxon rank-sum test. **F** Comparison of B cell clonal expansion estimated using bulk RNA-seq data, between responders (*n* = 15) and non-responders (*n* = 30). The *y*-axis represents the scaled Simpson index, where higher values indicate greater clonal expansion. Each dot represents an individual mouse. Statistical significance was determined using a one-tailed Wilcoxon rank-sum test. **G** AUC (Area Under the Curve) values for predicting ICB response based on the bulk B cell clonal expansion index, bulk T cell clonal expansion index, and single-cell $T_{em}$ cell clonal expansion index at Day 9. In box plots (**D**–**F**), the center line indicates the median; the box spans the interquartile range (IQR, 25th to 75th percentile); whiskers extend to values within 1.5× IQR from the quartiles; and each dot represents one biological replicate (a single mouse). Source data are provided as a Source Data file.

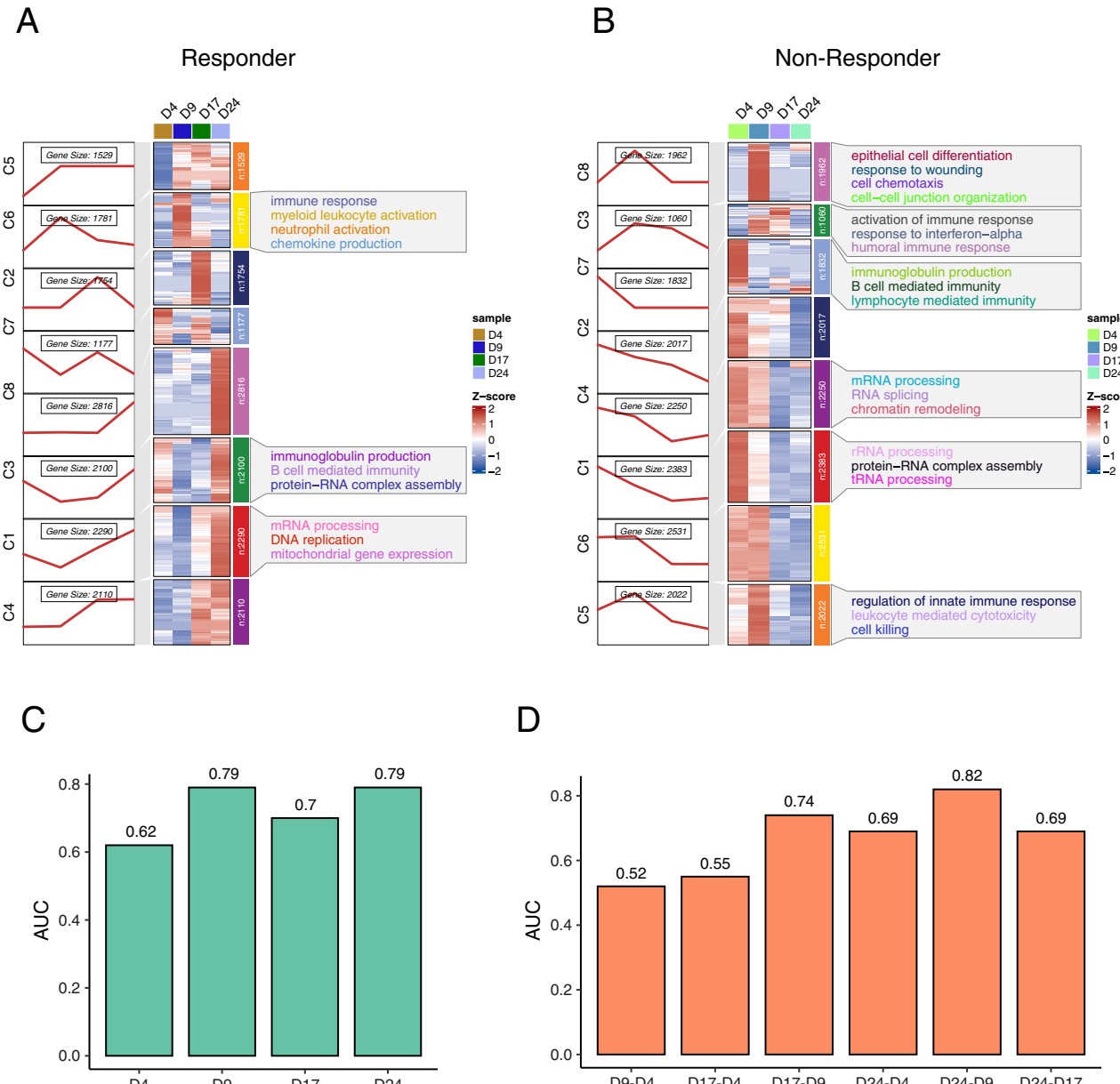

**Fig. 4 | Dynamic changes in gene expression following ICB treatment in blood.**
**A** Clusters of gene expression in responders. Each row of the heatmap represents one gene, and each column represents one time point. Colors indicate scaled gene expression levels. The line plot to the left of the heatmap shows the expression change pattern of each cluster. On the right side of the heatmap are the pathways enriched based on the genes within each corresponding cluster. **B** Clusters of gene expression in non-responders, with the same representation as in (**A**). **C** Machine learning models were trained for each time point using bulk RNA-seq data. The ICB response prediction performance of these time point-specific models was validated using five-fold cross validation. **D** Dynamic changes in gene expression between different time points were used to train a machine learning model. The ICB response prediction performance of this model was validated using five-fold cross validation. Source data are provided as a Source Data file.

points, especially in responders. To translate these findings into a clinical context and improve outcome prediction for ICB treatment, we identified $T_{em}$ and B cell gene response prediction signatures using single-cell expression data from the early on-treatment time point. An effector memory CD8+ T cell and B cell gene signature were identified, composed of the expression of 164 and 137 genes, respectively. Among the genes comprising the $T_{em}$ signature, several well-established markers are present, including *Cxcr3*, *Cxcr6*, and *Nkg7*. Interestingly, we also identified genes previously characterized as markers of transitional T cells (e.g., *Cx3cr1*, *Tbx21*, *Gzmb*) (SF 6A, B). These transitional T cells differentiate from stem-like Tcf-1+ CD8+ T cells and do not progress to an exhausted state[29,30]. These cells exhibit enhanced

functionality in T cell-mediated cytotoxicity and can be stimulated by anti-PD-1 inhibitors. This observation further supports the critical role of $T_{em}$ cells in mediating the response to ICB. In the B cell gene signature, canonical B cell markers such as *Cd19*, *Cd22*, *CD79a*, *CD79b*, and *Ms4a1* were identified. Additionally, novel genes not previously associated with these cell types were also identified (Supplementary Data 1).

To evaluate the predictive power of these mouse-derived signatures for predicting patients' response, we tested our predictive biomarkers in a total of seven ICB-treated HNSCC cohorts. These included four bulk RNA-seq datasets[31–34] and two single-cell datasets[35,36] and one new in-house ICB-treated blood single-cell

HNSCC dataset, previously unpublished. Both the $T_{em}$ and B cell signatures demonstrated robust performance in predicting ICB response in blood samples. Specifically, the $T_{em}$ signature achieved an average AUC of 0.69 ± 0.15, while the B cell signature achieved an average AUC of 0.74 ± 0.11 (Fig. 5A). To enhance accuracy and robustness, we combined the $T_{em}$ signature and the B cell signature into a composite score, termed the LiBIO (Liquid Biomaker of Immunotherapy Outcomes) score, by simply calculating their mean value in an unsupervised fashion. Notably, this LiBIO score demonstrated greater robustness and predictive accuracy compared to either signature alone, achieving a mean AUC of 0.80 ± 0.10 (Fig. 5A). To assess whether dynamic changes in blood reflect corresponding immune activity within the tumor microenvironment, we first generated in-house bulk RNA-seq data from tumors of HNSCC mice treated with anti-PD-1. Tumor biopsies were collected 14 days after cell implantation from five responders and four non-responders (SF 7A). We also analyzed an independent murine dataset in which single-cell RNA-seq and TCR-seq were performed on tumors from ICB-sensitive and ICB-resistant HNSCC models. Across both datasets, effector memory T cell, B cell, and LiBIO scores were elevated in ICB-sensitive or responder tumors compared to resistant or non-responder tumors (two-tailed Wilcoxon rank-sum test; SF 7B, C). While not statistically significant due to limited sample size, the trends were consistent and biologically meaningful. Notably, in the ICB-sensitive tumors, T cell clonal expansion significantly increased following treatment (two-tailed Wilcoxon rank-sum test; SF7 D), mirroring the dynamics observed in the peripheral blood. These results suggest that the clonal expansion detected in the blood is reflective of tumor-intrinsic immune activation and support the notion that blood-derived immune signatures, such as the LiBIO score, capture biologically meaningful tumor immune dynamics during ICB therapy.

We then turned to evaluate the $T_{em}$, B cell, and LiBIO scores in tumor datasets from HNSCC patients. Both the $T_{em}$ and B cell signatures performed well, with comparable predictive accuracy in bulk and scRNA-seq datasets. Specifically, the $T_{em}$ signature achieved an average AUC of 0.75 ± 0.087, and the B cell signature achieved an average AUC of 0.74 ± 0.12. Moreover, the LiBIO score outperformed both individual signatures, with an average AUC of 0.78 ± 0.046 (Fig. 5B). To benchmark the LiBIO score against FDA-approved biomarkers, we next compared it to the PD-L1 CPS. Due to the lack of publicly available ICB-treated HNSCC datasets with both transcriptomic profiles and clinical CPS annotations, we estimated CPS using a publicly available single-cell RNA-seq dataset ("Method") previously incorporated in this study[37]. This dataset contains both malignant and immune cells, allowing for transcriptomic estimation of CPS based on CD274 expression. The LiBIO score showed superior predictive performance compared to the estimated CPS score in Receiver Operating Characteristic (ROC) analysis (SF 8A, B). Nonetheless, because CPS was inferred from transcriptomic data rather than immunohistochemistry, further validation in datasets with clinically measured CPS is warranted. Subsequently, we assessed the predictive efficacy of the LiBIO score across patient cohorts, comparing its performance to several contemporary transcriptomics-based biomarkers[15,16,18,19,38–43]. Overall, the LiBIO score exhibited superior predictive performance relative to alternative biomarkers (Fig. 5C). Interestingly, two functional CD8+ T cell related tumor signatures also performed strongly in bulk RNA-seq cohorts, highlighting the pivotal role of functional CD8+ T cells in mediating ICB responses. However, while these functional CD8+ T cell related signatures performed well in bulk datasets, their performance was less consistent in single-cell cohorts. In contrast, the LiBIO score demonstrated greater robustness and predictive accuracy across both bulk and scRNA-seq datasets (Fig. 5C). Additionally, survival analyses revealed that patients with higher LiBIO scores had significantly improved survival outcomes compared to those with lower scores. This predictive association was more pronounced in ICB-treated cohorts and remained significant in ICB-treatment-naïve datasets (Fig. 5D) (SF 9A–C). Importantly, although the signatures were derived from an HPV-negative HNSCC mouse model, they remained significantly correlated with patient survival after adjusting for HPV status (Fig. 5D).

A universal fixed threshold could further facilitate the clinical utility of the LiBIO score for evaluating ICB response in HNSCC patients. Learning from a few cohorts used to optimize this decision threshold, we identified a LiBIO threshold of 0.424 that best distinguished responders from non-responders in single-cell datasets (SF 10A) and 0.201 in bulk datasets (SF 10B) ("Methods"). To understand the differing thresholds between bulk and single-cell data, we analyzed the distribution of LiBIO scores across both data types. Single-cell cohorts exhibited higher LiBIO scores compared to bulk RNA-seq cohorts (SF 11A). This was accompanied by significantly higher estimated abundances of B and T cells in the single-cell cohorts (SF 11B) ("Methods"), likely due to technical differences. These observations help explain why the single-cell threshold (0.424) is higher than the bulk RNA-seq threshold (0.201). Using this optimized decision threshold, the odds ratio for response in the single-cell training dataset was 4.0 (Fig. 5E) and 5.0 in the bulk dataset (Fig. 5F). Reassuringly, when applied to independent HNSCC patient datasets, the mean odds ratio was even higher, at 5.6 ± 2.1 in single-cell datasets (Fig. 5E) and 4.70 ± 2.5 in bulk datasets (Fig. 5F). These findings highlight the utility of $T_{em}$ and B cell signatures in predicting ICB response. Furthermore, a fixed LIBIO score threshold shows promise for clinical application, providing a robust and reliable method to distinguish responders from non-responders in HNSCC patients undergoing ICB treatment.

## Cross-cancer validation of the LiBIO score as a predictive biomarker for ICB response

$T_{em}$ and B cells play critical roles in mediating ICB responses, not only in HNSCC, but also across multiple cancer types[17,44–46]. To evaluate the broader applicability of the LiBIO score beyond HNSCC, we analyzed its predictive performance across 11 ICB-treated patient groups, including both pre-treatment and on-treatment samples, from four melanoma cohorts[4,47–49], three NSCLC cohorts[50–52], and two breast cancer cohorts[53,54]. The LiBIO score achieved an average AUC of 0.80 ± 0.09 in melanoma, 0.73 ± 0.23 in NSCLC, and 0.72 ± 0.10 in breast cancer cohorts (Fig. 6A). To further enhance clinical interpretability, we aimed to identify fixed decision thresholds for LiBIO score classification specific to each cancer type. For each data type, one cohort was designated as a training cohort to determine the optimal threshold that maximized the odds ratio for ICB response. The optimal thresholds identified were 0.196 for melanoma, 0.632 for NSCLC, and 0.383 for breast cancer (SF 12A–C). In the respective training cohorts, these thresholds achieved odds ratios of 4.7, 1.4, and 2.1 (Fig. 6B). We then applied the same thresholds to the remaining cohorts of each cancer type to evaluate generalizability. In these testing cohorts, LiBIO maintained strong performance, achieving average odds ratios of 3.4 ± 0.34 in melanoma (excluding the Riaz et al. cohort due to RNA-later biopsy preparation, which differs from FFPE used in the other cohorts), 2.3 ± 0.14 in NSCLC, and 1.3 in breast cancer (Fig. 6B). These results highlight the potential of the LiBIO score as a robust and clinically applicable biomarker for predicting ICB response across multiple tumor types.

## Discussion

The host antitumor immune response is both dynamic and coordinated, characterized by the trafficking of immune cells from the tumor microenvironment to tumor-draining lymph nodes and subsequently into systemic circulation. This intricate yet synchronized immunobiology supports longitudinal peripheral sampling as a means to monitor the overall host immune response and serve as a surrogate for predicting tumor-specific immunotherapy outcomes. Here, we systematically characterized dynamic changes in the blood during ICB

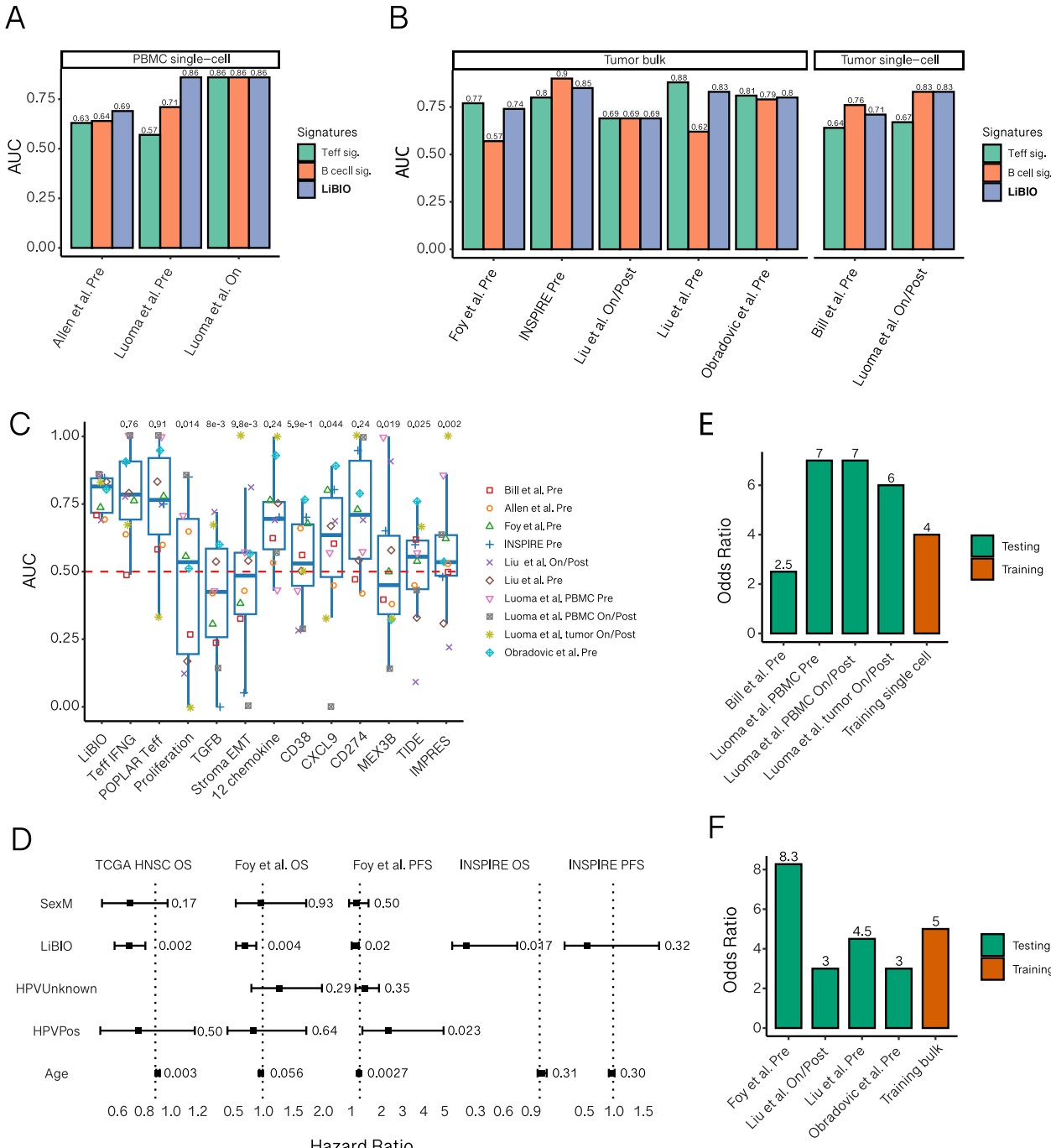

**Fig. 5 | T<sub>em</sub> and B cell signatures predict ICB response in human patients.**

**A**, **B** AUC (Area Under the Curve) values for ICB response prediction based on Tem and B cell signature scores, as well as the combined score (calculated as the mean of Tem and B cell signature scores) in both blood (**A**) and tumor (**B**) HNSCC cohorts. **C** AUC values for the combined score compared to previously published transcriptomic signatures across 10 HNSCC cohorts. Each dot represents one HNSCC cohort, displayed using different colors and shapes. The box plot displays the median (center line), interquartile range (IQR; box limits: 25th to 75th percentile), and whiskers extending to 1.5× IQR from the quartiles. Two-tailed P-values were calculated using the Wilcoxon rank-sum test to compare the LiBIO score against

other signatures. **D** Hazard ratios (HRs) for overall survival per 1-unit increase in the combined T$_{em}$ and B cell score (LiBIO score), adjusted for age and sex, in three independent HNSCC cohorts: TCGA ($n = 516$), Foy et al. ($n = 102$), and INSPIRE ($n = 12$). Dots represent HR estimates; error bars indicate the 95% confidence interval (CI). Statistical significance was assessed using the Wald test.
**E**, **F** Identification of fixed thresholds for the LIBIO score in single-cell (**E**) and bulk (**F**) cohorts. The X-axis represents the cohorts, while the Y-axis indicates the odds ratio (OR) of responders versus non-responders (see "Methods"). Orange bars correspond to training cohorts, and green bars represent independent validation cohorts. Source data are provided as a Source Data file.

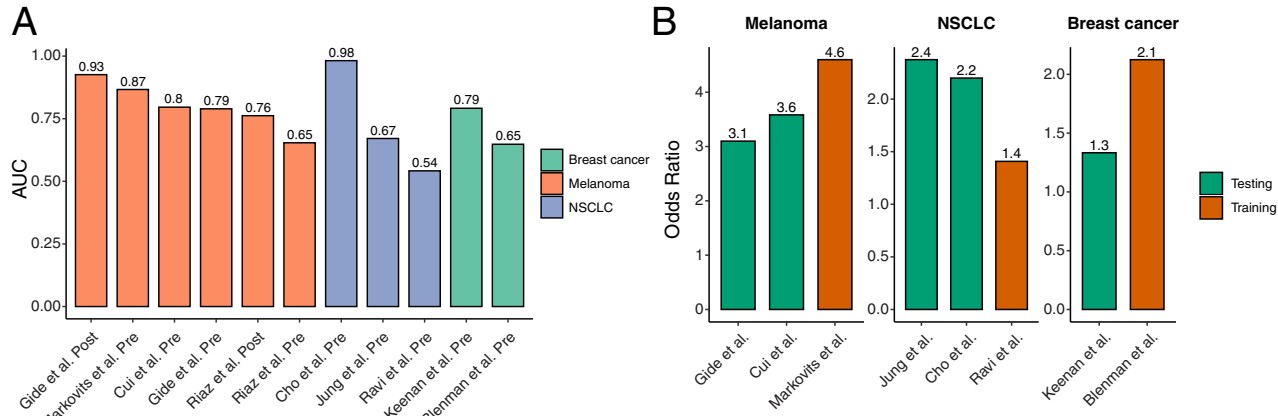

**Fig. 6 | Prediction of immune checkpoint blockade (ICB) response using the LiBIO score across multiple cancer types. A** Area under the curve (AUC) values for LiBIO score-based prediction of ICB response across 11 patient groups. The *x*-axis labels indicate both the dataset source and the timing of sample collection relative to anti-PD-1 therapy, where "Pre" denotes samples collected before treatment initiation and "Post" denotes samples collected after treatment had begun.

**B** Odds ratio (OR) for distinguishing responders from non-responders using fixed LiBIO thresholds specific to melanoma, NSCLC, and breast cancer (see "Methods"). Orange bars represent training cohorts used to determine the optimal threshold, and green bars correspond to independent validation cohorts. Source data are provided as a Source Data file.

treatment across different time points in a mouse model of head and neck cancer. This approach establishes a platform for developing high-fidelity, non-invasive liquid biomarkers that enable real-time prediction and monitoring of immunotherapy outcomes. Consistent with this, our preclinical findings highlight the potential of longitudinal liquid biopsy approaches to track immune dynamics. Serial blood draws from oral cavity tumor-bearing mice treated with PD-1 blockade revealed an early, transient expansion of effector memory T ($T_{em}$) and B cell repertoires, preceding tumor regression. These findings underscore the pivotal role of $T_{em}$ and B cells in mediating ICB responses in head and neck cancer[7,28]. Furthermore, temporal shifts in clonality and gene expression at early treatment time points emerged as strong predictors of ICB response, identifying these windows as optimal for assessment. Notably, the translation of a newly identified combined immune effector signature termed LiBIO to human cohorts demonstrated its broad applicability in predicting ICI outcomes, motivating its further prospective study. The LiBIO score not only demonstrates predictive utility for ICB response in both blood and tumor transcriptomic datasets but also exhibits prognostic value in both ICB-treated and ICB-naïve cohorts. While this suggests that LiBIO reflects core features of anti-tumor immune activation, future studies are needed to disentangle its predictive versus prognostic roles in distinct clinical contexts.

As with any study, our work has a few limitations. First and foremost, analogous time series data from tumors or blood of human HNSCC patients does not yet exist, which restricted our ability to directly compare the temporal dynamics observed in our mouse findings to human data. Additionally, we could only find one publicly available HNSCC patient cohort contains PBMC data, which could enable us to test our blood generated signature directly (at the pre-treatment time point). To partially overcome this challenge, we performed an additional blood-based analysis in a small in-house cohort we have generated. Due to the scarcity of PBMC datasets, we extended our validation to tumor-derived datasets. While several tumor datasets are available, most still have small sample sizes. Fortunately, two tumor cohorts with larger sample sizes were accessible, providing a more robust validation of the identified signatures. Another limitation to note is that single-cell TCR sequencing data can only capture a few hundred T cells for each sample, limiting the comprehensiveness of TCR tracing. This constraint hindered our ability to identify specific T cell clones associated with ICB response in our data.

We identified five days after ICB treatment as the early on-treatment time point in mouse that is optimal for assessing ICB response in mouse. This strategy may also be applicable for HNSCC patients, but obviously the best timing of such early on-treatment time point in humans requires further careful investigation. The resulting LiBIO score integrates the $T_{em}$ and B cell signatures, and quite notably, its application using one fixed decision threshold yields a fairly high predictive odds ratio of response, which surpasses the predictive power of existing biomarkers and a transcriptomic based CPS score. Although this finding is encouraging, we emphasize that the CPS score was derived from RNA expression data and not standard clinical IHC measurements. Collectively, our preclinical insights support the premise that peripheral immune events can serve as a foundation for biomarker discovery, offering a non-invasive, biologically grounded approach to monitor and predict immunotherapy outcomes with high-fidelity. Finally, the approach presented lays a solid basis for developing similar biomarkers in other cancer indications.

## Methods

All animal procedures were conducted in accordance with relevant ethical regulations and were approved by the Institutional Animal Care and Use Committee at the University of California, San Diego (UCSD) under protocol number S16200. The approved protocol, titled "Mouse model for cancer development and drug treatment of cancer", was granted on August 23, 2022, and remains valid through July 21, 2025. It covers the use of Mus musculus (mouse) in full compliance with federal and institutional guidelines.

### Mouse and anti-PD-1 treatment

C57BL/6J mice (6–8 weeks old) were housed under specific pathogen-free conditions and used in compliance with institutional animal care guidelines. Tumors were established by injecting 4MOSC1 cells ($1 \times 10^5$) into the tongue or buccal mucosa, as previously described. Mice were treated intraperitoneally with αPD-1 (clone RMP1-14, Bio × Cell, 200 µg) every three days starting on day 7 on-tumor implantation. Mice were evaluated at least three times per week to monitor tumor progression, weight, grooming, and general condition. Tumor size was recorded using calipers. Studies were concluded at predetermined endpoints or earlier if animals showed signs warranting humane euthanasia, such as >20% body weight loss, inability to groom or ambulate, distress, or tumor ulceration. According to institutional

animal care guidelines, mice were euthanized if tongue tumors exceeded 8 mm or buccal tumors exceeded 10 mm, or in the case of ulceration. Both male and female mice were used in the study. Data were disaggregated by sex where relevant, and sex was considered during study design to evaluate any potential sex-based differences. No significant sex-based differences were observed; therefore, combined data are presented.

## Sample collection

Mice were bled via retro-orbital puncture using heparinized EDTA-coated glass capillary tubes. Blood was immediately transferred into microcentrifuge tubes containing TRIzol reagent (Thermo Fisher Scientific) for RNA stabilization. Samples were processed according to the manufacturer's instructions for RNA extraction. For scRNA-seq, blood was diluted in PBS + 0.04% BSA, and cell concentration and viability were determined using the Countess II Automated Cell Counter, targeting 700–1200 cells/μL for downstream analysis.

## In-house HNSCC patient single-cell ICB cohort

Twenty pre-treatment PBMC samples were obtained from patients with newly diagnosed advanced-stage HPV-negative oral cavity cancers enrolled in a neoadjuvant immunotherapy clinical study. Deidentified PBMC were used for experimental purposes after patients provided full informed consent under NIH Biospecimen Protocol Number 18-DC-0051 (NCT03429036). The therapeutic agent used was bintrafusp alfa, a dual PD-L1 and TGF-β blocker. As part of this neoadjuvant study, clinical responses were assessed based on pathological responses, specifically by measuring tumor regression in the surgical specimens to determine the degree of tumor shrinkage following immunotherapy. Clinical response was treated as a continuous variable, represented by the percentage of tumor shrinkage. To classify patients into responder and non-responder groups, an arbitrary cutoff was applied. Samples with tumor shrinkage greater than 50% were categorized as responders, while those with 50% or less tumor shrinkage were categorized as non-responders.

## Public HNSCC patient cohorts

Four publicly available bulk RNA-seq, Foy et al.[31], INSPIRE[34], Liu et al.[33], Obradovic et al.[32], and two single-cell RNA-seq, Bill et al.[36], Luoma et al.[35], datasets from ICB-treated HNSCC patients were collected. For the bulk RNA-seq data, Transcripts Per Million (TPM) values were obtained from the original publications. For the single-cell datasets, raw counts and cell annotation information were retrieved from the respective original studies. Data normalization and pseudobulk analysis for the single-cell data were performed using the NormalizeData and AverageExpression functions from the Seurat package[55]. Clinical information, including ICB response status and patient survival data, were also collected from the original publications. Response status of HNSCC patients was based on RECIST criteria[56], with "CR/PR" patients classified as responders and "SD/PD" patients classified as non-responders.

## Single-cell RNAseq data analysis

Raw scRNA-seq reads were barcode-deduplicated and aligned to the mm10 reference genome using Cell Ranger[57] to generate count matrices. These count matrices were then used as input for Seurat to identify cell types and cellular states. Cells with more than 25% mitochondrial content or fewer than 500 expressed genes were removed from downstream analysis. Genes expressed in fewer than 3 cells were also excluded. Variable feature identification was performed using the FindVariableFeatures function with a parameter of 2000 features, followed by clustering with the FindClusters function. Marker genes for each cluster were identified using the FindMarkers function, and major cell types were annotated based on feature genes. Pseudobulk gene expression was calculated using the AverageExpression function.

CD8+ and CD4+ T cells were subset based on the expression of Cd8a, Cd8b1, and Cd4 using the scGate package[58]. Specifically, CD8+ T cells were identified using the signature Cd8a+, Cd8b1+, Cd4−, and CD4+ T cells using the signature Cd8a−, Cd8b1−, Cd4+. CD8+ and CD4+ T cells were then mapped to a mouse T cell database to annotate their subtypes using ProjecTILs[59].

## TCR and BCR repertoire analysis

Single-cell TCR sequencing data were aligned to the mouse mm10 V(D)J reference genome using Cell Ranger to obtain clonotype information for each cell. The filtered contig annotation file from the Cell Ranger output was used for downstream analysis. Bulk RNA-seq data were also used to identify T cell and B cell clones using TRUST4[60]. Clone size was calculated by counting the number of identical clones in each sample. Clonal expansion was measured using the Simpson Index, calculated with the immunarch package[61].

## Bulk RNAseq data analysis

Mouse bulk RNA-seq data were first aligned to the reference genome mm10 to obtain raw counts. TPM values were then calculated based on raw counts and reference genome information. Genes were clustered using the fuzzy c-means algorithm[62] from the e1071 package[63]. The optimal number of clusters was determined using the Elbow method. Gene functional annotation was performed using clusterProfiler[64] and data visualization were performed using ComplexHeatmap[65].

## Identification of $T_{em}$ and B cell signatures and signature score calculation

Cells from Day 9 samples were subset from the full dataset. The FindMarkers function from Seurat package was then applied to these subset cells to identify marker genes for each cell type. Marker genes for $T_{em}$ and B cells were defined based on a false discovery rate (FDR ≤ 0.01) and fold changes (≥ 1.5). The signatures generated from the mouse single-cell data were then mapped to human gene symbols to ensure compatibility with human datasets. Genes not expressed in at least one of the six HNSCC cohorts were filtered out. The $T_{em}$ and B cell signature scores for each sample were calculated using the ssGSEA algorithm[66]. The combined $T_{em}$ and B cell score was computed as the mean value of the $T_{em}$ and B cell scores.

## Estimation of immune cell abundance

To estimate immune cell abundance across different data modalities, we applied complementary approaches tailored to single-cell and bulk RNA-seq datasets.

For single-cell RNA-seq cohorts, immune cell abundance was calculated directly based on cell type annotations. Specifically, we computed the relative abundance of B cells and T cells by dividing the number of annotated cells of each type by the total number of cells within each sample. Cell annotations were assigned using canonical markers and validated clustering from the original publications or our in-house pipeline.

For bulk RNA-seq cohorts, we estimated immune cell abundance using CODEFACS (COnfident DEconvolution For All Cell Subsets)[67], a robust deconvolution framework that enables accurate inference of cell type-specific signals from bulk RNA-seq data. As the input reference signature, we used a curated single-cell RNA-seq dataset[36] from HNSCC, which includes well-characterized malignant, immune, and stromal cell populations. This reference was used to deconvolve bulk expression profiles and extract relative abundance estimates for B cells and T cells across samples. All deconvolution steps were performed using default parameters unless otherwise specified.

## Calculation of PD-L1 combined positive score from single-cell RNA-seq data

To approximate the clinically defined PD-L1 CPS using scRNA-seq data, we computed a transcriptomic analog based on the expression of *CD274*, which encodes PD-L1.

Cell annotations were obtained from the original publication and used to classify each cell as either a tumor cell or an immune cell. Non-relevant cell types (e.g., stromal or endothelial cells) were excluded from downstream analysis.

A cell was considered PD-L1 positive if more than two reads were mapped to the *CD274* gene. This threshold was chosen to minimize potential noise from low-level or spurious expression often observed in single-cell RNA-seq data. For each sample, the CPS was calculated using the following formula:

$$CPS = \frac{Number\ of\ PDL1^+\ tumor\ cells + Number\ of\ PDL1^+\ immune\ cells}{Total\ number\ of\ tumor\ cells} \times 100 \tag{1}$$

## ICB prediction performance evaluation and the determination of threshold of combined $T_{em}$ and B cell score

The ICB response prediction performance was measured using the Area Under the Receiver Operating Characteristic (ROC) Curve (AUC) and the Odds Ratio (OR) of responders to non-responders. AUC is a standard metric in machine learning that evaluates the overall predictive performance of a classifier across all possible decision thresholds. The OR represents the odds of responding when the treatment is recommended, divided by the odds of responding when the treatment is not recommended. It quantifies performance at a specific decision threshold, making it a more clinically relevant measure. The detailed calculations for these two metrics are provided below:

The AUC is defined as the area under the ROC curve, which plots the true positive rate (sensitivity) against the false positive rate (1-specificity) at various threshold settings. The AUC is calculated using the following equation:

$$\int_0^1 TPR(FPR)d(FPR) \tag{2}$$

Where:
- TPR is the True Positive Rate (sensitivity), calculated as $\frac{TP}{TP+FN}$.
- FPR is the False Positive Rate (1-specificity), calculated as $\frac{FP}{FP+TN}$.

AUC values range from 0 to 1, where 1 indicates perfect model performance, and 0.5 indicates random chance.

The odds ratio (OR) is a measure of association between exposure (in this case, the treatment recommendation) and outcome (response to ICB treatment). It is defined as the odds of responding to treatment when it is recommended, divided by the odds of responding when it is not recommended. The OR is calculated using the following equation:

$$OR = \frac{\frac{TP}{FP}}{\frac{FN}{TN}} = \frac{TP \times TN}{FP \times FN} \tag{3}$$

Where:
- TP (True Positive) is the number of responders correctly identified as responders.
- FP (False Positive) is the number of non-responders incorrectly identified as responders.
- TN (True Negative) is the number of non-responders correctly identified as non-responders.
- FN (False Negative) is the number of responders incorrectly identified as non-responders.

An OR greater than 1 indicates that the treatment increases the likelihood of response, while an OR less than 1 suggests that the treatment decreases the likelihood of response. An OR of 1 means the treatment has no effect on the odds of response.

## Survival analysis

Cox proportional hazards regression[68] was used to assess the association between gene signatures and interaction score and patient survival with age and sex as covariant. This model estimates the hazard ratio (HR) for each covariate, which represents the relative risk of an event (e.g., death or progression) occurring at any given time. Hazard ratios greater than 1 indicate increased risk, while values less than 1 suggest decreased risk. Kaplan–Meier survival curves were generated to estimate survival probabilities and visualize differences between high score group and low score group (samples were divided by the mean of score). Log-rank tests were used to compare survival distributions between the groups, with *P*-values indicating whether the differences in survival were statistically significant.

## Statistics & reproducibility

Sample size was determined based on prior studies and statistical power calculations to ensure sufficient power to detect significant effects. No data exclusions were performed; all data points were included in the analysis. All experiments were replicated at least twice with consistent results, ensuring reproducibility. Mice were randomly assigned to experimental groups to control for potential confounding variables. Investigators were blinded to group allocation during data collection and analysis to minimize bias.

Unless otherwise stated, a one-tailed Wilcoxon rank-sum test[69] was used to assess differences in distributions between two population groups. Odds ratios were calculated using Fisher's exact test[55]. All statistical analyses were conducted using R version 4.4.1[70].

## Reporting summary

Further information on research design is available in the Nature Portfolio Reporting Summary linked to this article.

## Data availability

The bulk RNA sequencing data generated in this study are publicly available in the NCBI Gene Expression Omnibus (GEO) under accession code GSE299686. The single-cell RNA sequencing (scRNA-seq) and single-cell T cell receptor sequencing (scTCR-seq) data are publicly available under accession code GSE299683. Source data supporting the findings of this study are provided with this paper. Source data are provided with this paper.

## Code availability

All original code used in this study has been deposited in GitHub at https://github.com/wbb1813/Time_series_mouse_ICB and is publicly available as of the date of publication. To ensure reproducibility and provide a permanent citation, the repository has also been archived in Zenodo with the https://doi.org/10.5281/zenodo.15856815[71].

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

## Acknowledgements
This research is supported in part by the Intramural Research Program of the NIH, NCI, Center for Cancer Research. This work utilized the computational resources of the NIH HPC Biowulf cluster.

## Author contributions
B.W. and K.W. conducted computational analyses with assistance from S.S., S.M., S.P., S.R.D., D.W.; R.S.K., L.C., and S.T. performed the mouse experiments; X.Y. and C.A. generated the patient single-cell data. B.W. drafted the original manuscript, while P.A.D., K.W., S.R.D., J.S.G., R.S.K., and E.R. contributed to revisions. All authors reviewed and approved the final manuscript.

## Funding

## Competing interests
E.R. is a co-founder of Medaware Ltd. (https://www.medaware.com/), Metabomed (https://www.metabomed.com/), and Pangea Biomed (https://pangeamedicine.com/). He has divested and serves as an unpaid scientific consultant to the latter company. J.S.G. is a consultant/advisory board member for Pangea Biomed, Radionetics, and io9, and founder of Kadima Pharmaceuticals. The rest of the authors declare no conflicts of interest.

## Additional information

[1]Cancer Data Science Laboratory, Center for Cancer Research (CCR), National Cancer Institute (NCI), National Institutes of Health (NIH), Bethesda, MD, USA. [2]Department of Otolaryngology-Head and Neck Surgery, UC San Diego School of Medicine, San Diego, CA, USA. [3]Moores Cancer Center, UC San Diego, La Jolla, CA, USA. [4]Gleiberman Head and Neck Cancer Center, UC San Diego, La Jolla, CA, USA. [5]Department of Head and Neck Surgery, The University of Texas MD Anderson Cancer Center, Houston, TX, USA. [6]Department of Engineering, Stanford University, San Jose, CA, USA. [7]Department of Pathobiology, University of Illinois Urbana-Champaign, Urbana, IL, USA. [8]Surgical Oncology Program, Center for Cancer Research (CCR), National Cancer Institute (NCI), National Institutes of Health (NIH), Bethesda, MD, USA. [9]Department of Hematology & Medical Oncology, Fox Chase Cancer Center, Philadelphia, PA, USA. [10]Department of Comparative Biosciences, University of Illinois Urbana-Champaign, Urbana, IL, USA. [11]Cancer Center at Illinois, University of Illinois Urbana-Champaign, Urbana, IL, USA. [12]Department of Bioengineering, University of Illinois Urbana-Champaign, Urbana, IL, USA. [13]Department of Pharmacology, UC San Diego, La Jolla, CA, USA. [14]These authors contributed equally: Binbin Wang, Robert Saddawi-Konefka. ✉e-mail: kwang222@illinois.edu; sgutkind@health.ucsd.edu; eytan.ruppin@nih.gov

