## [Transparent Peer Review file · Nature Communications]

Longitudinal liquid biopsy identifies an early predictive biomarker of immune checkpoint blockade response in head and neck squamous cell carcinoma

Corresponding Author: Dr Eytan Ruppin

Version 0:

Reviewer comments:

Reviewer #1

(Remarks to the Author)

This study investigates the use of longitudinal liquid biopsies to identify predictive biomarkers for immune checkpoint blockade (ICB) response in head and neck squamous cell carcinoma (HNSCC). Using a syngeneic mouse model, the authors profile peripheral blood mononuclear cells (PBMCs) at four time points through bulk and single cell RNA sequencing as well as T cell receptor sequencing. They identify early clonal expansion of effector memory CD8 positive T cells and B cells in responders, leading to the development of the LiBIO score, a composite biomarker that integrates transcriptional signatures from both cell types. The LiBIO score demonstrates predictive utility across multiple HNSCC cohorts and appears to perform better than existing biomarkers. The preclinical design is rigorous and the translational relevance is evident. Specific comments are outlined below:

1. The central premise of the study suggests that early treatment-related immune changes in blood can predict ICB response, which is well supported by the murine data. However, validation in human cohorts remains limited due to the absence of corresponding temporal information. The predictive value of the LiBIO score in human studies is inferred from static pre-treatment samples rather than dynamic immune changes during therapy, which limits direct translational applicability. In addition, most human data are derived from tumor tissue rather than blood. Validating the effectiveness of the liquid biopsy derived score in murine tumor samples would help strengthen the comparison and enhance translational relevance.
2. The study focuses specifically on ICB response in HNSCC, which is clinically appropriate. However, it would be informative to examine whether the LiBIO score is applicable to other tumor types using publicly available datasets. A discussion on whether the immune landscape during ICB treatment in HNSCC is distinct from that in other cancers would help clarify whether the score has tumor-type specificity or broader utility.
3. The LiBIO score applies different thresholds for bulk RNA sequencing (0.201) and single cell RNA sequencing (0.424), which may complicate its use in clinical settings. It is important to clarify whether the lower threshold in bulk RNA data reflects technical noise or dilution of immune signals due to the presence of non-immune cells. Whether the expanding cohort will influence the threshold should also be considered. A percentile-based threshold may offer greater robustness and improve generalizability across patient populations.
4. The predictive performance of the LiBIO score should be compared directly with established clinical biomarkers in HNSCC, such as the PD-L1 combined positive score (CPS). The absence of CPS in the benchmarking analysis reduces the translational impact of the study. Including CPS and exploring whether a combined biomarker strategy improves predictive accuracy would strengthen the clinical relevance of the findings.

(Remarks on code availability)

Reviewer #2

(Remarks to the Author)

In this interestingly titled study by Wang et al, the authors use a previously established syngeneic mouse model of HNSCC, treat these with anti-PD1 blockade and obtain blood from these mice at 4 different time points to determine: bulk RNAseq, scRNAseq and scTCRseq from PBMCs. The main objective here is to establish the dynamics of PMBC comparing responders with non-responder mice. This is a nice system as it appears that a third of the mice appear to respond to the treatment while 2/3rds do not, but little detail is provided on the mouse system used. I had to trawl through this and previous papers to even understand the model, and I am uncertain given the single genetic background of a cell line and the mouse, how clonality varies between the responders and non-responders to account for this differential response. Furthermore, I am also concerned that the entire basis of this paper is on a single mouse line (albeit very comprehensively analysed), and I feel that further validation on a different system would be critical, especially since this data covers 4/5 of the figures in the paper; at the very least the critical parts of the study, time points and validation of the Tem, B and clonal dynamics that are presented here.

The other concern I have with this model is that a 'clinical' response is apparent after 10 days, which is of course sooner than what I would expect for a human patient (which makes this model attractive), BUT poses a problem when trying to compare the dynamics here to human timelines. Furthermore, any biomarker AFTER clinical response is pretty useless as well, so data from timepoints 4 is AFTER a clinical response and hence not particularly relevant. Timepoint 2 is therefore the more interesting timepoints to focus on here. would be great if all the plots in figures 1 and 2 could be labeled by actual days of treatment rather than time points 1,2,3,4, which is harder to follow.

Other critical points to be addressed are as follows:

1. scRNAseq supports the role of Tem dynamics in overall response and enhanced in the responder, but the role of B-cell activation and the dynamics presented are statistically weak and is a bit of an over claim here.
2. Throughout the paper, the authors have chosen statistically weaker tests (like one tailed tests etc), and I wonder if this is because more robust test would weaken their 'p values'
3. the described methodology for scRNAseq and scTCRseq is completely wrong- how could they align their mouse data to human hg38 reference genome for RNAseq or TCRseq?
4. for the TCRseq, the results seem to favour BCRseq as a biomarker, especially in bulk sequencing...would be great if the authors expanded on this or explained it
5. As stated above, showing a good AUC after seeing a clinical response (timeline 4) is not useful, and neither is the dynamics between T4-T2. I would focus more on T2 as this pre-dates any clinical response.
6. Validation was done on data and timepoints that were available, and likely could have missed all the early timelines here. Also I would have liked to have seen PBMC TCR and BCR dynamics validation. Do the numbers really go up and prune off as quickly as suggested here?
7. Trying to validate the nice blood based biomarker on tumor datasets and TCGA (to correlate with OS), is a bit regressive. The validation should be like for like. The latter data just suggests to me that you have an all-encompassing immune marker that is overall prognostic rather than predictive. If there is no difference to other scoring system, it should be stated as such. Similarly, I do. not understand the continual used of a single-tail statistical test. Sounds like you have already decided your model is better!
8. The discussion should focus on comparing these markers to other established markers of PD1 response, some of which the senior author has co-written and shows much better AUC than this study.

(Remarks on code availability)

Reviewer #3

(Remarks to the Author)

The paper aims to look at biomarkers in a mouse model of HNSCC to determine if there is a pattern of changes in the blood that can determine whether a patient will elicit a positive response to the administration of immune check point blockade (ICB). The authors utilize single cell and bulk mRNA signatures.

The authors find differences in clonal expansion and temporal dynamics of effector T cells and B cells between responder and non-responder mice. These changes were seen early in treatment A LiBIO score (Liquid Biomarker of Immunotherapy Outcome) from these data predict ICB response in HNSCC patients. The authors propose these methods as a non-invasive, biologically grounded approach that can predict ICB outcomes with high fidelity. They suggest this method can be used to determine biomarkers for other cancer types.

The authors use a 4MOSC cell line, a carcinogen induced orthotopic tumor HNSCC model where tumors in c57bl6 mice were induced by 4-nitroquinoline-1 oxide (4NQO), implanted into the tongue (1x10⁵ cells). Blood was drawn at days 4, 9, 17- and 24-days post implant. PD-1 treatment was started immediately after the 4 day collection. Treatment efficacy was determined by tumor growth and immune responses. Bulk RNA-seq, scRNA-seq and scTCR sequencing was utilized to examine immune repertoire changes.

The authors created 2 data sets of 45, 16 mice cohorts with the sets split into responders and non-responders. They also incorporated 20 HNSCC patient samples from ICB trials. They also accessed two previously published ICB treatment data sets and 4 publicly available ICB bulk RNA-seq data sets.

The in-vivo murine data showed an increase in CD4, Cd8 and B cell abundance after treatment compared to pre-ICB and a loss of neutrophils. When examining subsets of the increased cells effector memory cells Tem and Th1 cells were increased in each time point after ICB compared to pre-treatment which was not seen in non-responders. A kinetic

difference in the accumulation of B-cells was also observed.

Clonal expansion of T and B cells were increased in responders after ICB compared to non-responders. Tem cells has a population that increased post ICB that was not seen in non-responders. This was validated utilizing a different cohort and using bulk RNAseq instead of scRNAseq showing T-cell expansion occurred early.

These early effects were predictive of response, with Bcells showing predictive efficacy at the 3 earliest time points (1,2,3). The changes in biological functions were assessed by algorithmic determination of 8 clusters of genes that are differentially regulated in the responders (5 upregulated compared to pretreatment including B-cell activation, immunoglobulin production and mitochondrial gene expression) and downregulated in the non-responders (6 down regulated including bcell activation and immunoglobulin production). Non-responders showed downregulation in innate immune response, leukocyte-mediated cytotoxicity, and cell killing were downregulated.

The authors generated a gene signature for CD8+ t cells (184 genes) and B cells (168 genes) and utilized the signatures for their predictive measure testing them against 7 ICB treated cohorts. The authors find significance in predicting the ICB response with AUC values of 0.96 and 0.74 for T cell and B cell clusters, respectively. They combine the signatures into a LiBIO score that improved the AUC value to 0.8.

These predictive tests were performed on human databases finding AUC values similar to the mouse data of 0.75, 0.74 and 0.78 for the T cell, B cell and combined signatures. This was summarized to be an improvement on other biomarkers.

Comments

It's an in-depth study of the effect of ICB on the immune abundance in patient blood before and after treatment. The findings are exciting from a diagnostic angle but also eludes to the differences in patients that respond to ICB highlighting the importance of T cell effector cells and B cells in HNSCC.

The study appears to be run in a well-designed fashion with the authors utilizing several different but parallel cohorts to determine the role of the cell signatures.

The authors acknowledge the limitations of the study; lack of analogous time series data from human patients, very few studies with PBMC data and small sample sizes.

The paper is well written with very few errors in grammar or spelling.

(Remarks on code availability)

Comments

It's an in-depth study of the effect of ICB on the immune abundance in patient blood before and after treatment. The findings are exciting from a diagnostic angle but also eludes to the differences in patients that respond to ICB highlighting the importance of T cell effector cells and B cells in HNSCC.

The study appears to be run in a well-designed fashion with the authors utilizing several different but parallel cohorts to determine the role of the cell signatures.

The authors acknowledge the limitations of the study; lack of analogous time series data from human patients, very few studies with PBMC data and small sample sizes.

The paper is well written with very few errors in grammar or spelling.

Version 1:

Reviewer comments:

Reviewer #1

(Remarks to the Author)

The revised manuscript adequately addresses the questions raised by the reviewers with strong additional data. The current version is more solid and concrete. I have no further comments on the manuscript, and I consider it acceptable for publication.

(Remarks on code availability)

Reviewer #2

(Remarks to the Author)

Thank you for the revised manuscript, addressing my comments and newer data added. I appreciate the addition of a second mouse model to validate the data and additional human data beyond HNSCC, where the Libio scoring system seems to hold true as well. There are some minor errors in labelling, colour and legends that should be corrected as well prior to publication (ugh app figure 7a, where the legend says red and blue lines, while the figure is blue and yellow)..these are minor but it is good to get these right, because a lot of data is in the supplementary section which is useful as a resource for other studies. Apart from that I support the publication of this manuscript. Congratulations to all the investigators...

(Remarks on code availability)

Reviewer #3

(Remarks to the Author)

Thank you for your response to our comments.

(Remarks on code availability)

I do not know how to interpret any of this code

Responses to reviewer comments

Within this document, the reviewer comments are copied verbatim in black font and alternate with our responses, which are presented in red font. Any text additions made to the manuscript to address the reviewers' comments are highlighted in turquoise and are indented, except for very short additions.

Reviewer #1 (Remarks to the Author): expertise in HNSCC immune checkpoint inhibition

This study investigates the use of longitudinal liquid biopsies to identify predictive biomarkers for immune checkpoint blockade (ICB) response in head and neck squamous cell carcinoma (HNSCC). Using a syngeneic mouse model, the authors profile peripheral blood mononuclear cells (PBMCs) at four time points through bulk and single cell RNA sequencing as well as T cell receptor sequencing. They identify early clonal expansion of effector memory CD8 positive T cells and B cells in responders, leading to the development of the LiBIO score, a composite biomarker that integrates transcriptional signatures from both cell types. The LiBIO score demonstrates predictive utility across multiple HNSCC cohorts and appears to perform better than existing biomarkers. The preclinical design is rigorous and the translational relevance is evident. Specific comments are outlined below:

1. The central premise of the study suggests that early treatment-related immune changes in blood can predict ICB response, which is well supported by the murine data. However, validation in human cohorts remains limited due to the absence of corresponding temporal information. The predictive value of the LiBIO score in human studies is inferred from static pre-treatment samples rather than dynamic immune changes during therapy, which limits direct translational applicability. In addition, most human data are derived from tumor tissue rather than blood. Validating the effectiveness of the liquid biopsy derived score in murine tumor samples would help strengthen the comparison and enhance translational relevance.

Response:

We appreciate the reviewer's insightful suggestion. To evaluate whether the blood-derived LiBIO score captures immune activity within the tumor microenvironment, we performed two complementary analyses using murine tumor samples. First, we generated in-house bulk RNA-seq data from tumors of HNSCC mouse models treated with anti-PD-1. Tumor biopsies were collected 14 days after tumor cell implantation, including samples from five responders and four non-responders (Figure R1A; Supplementary Figure 7A in the manuscript). Second, we analyzed a publicly available dataset of HNSCC mouse tumors in which both single-cell RNA-seq and single-cell TCR-seq were performed in ICB-sensitive (MOC1) and ICB-resistant (MOCE1) models¹. We calculated the effector memory CD8⁺ T cell (T_{em}), B cell, and LiBIO scores in both datasets. All three scores were consistently higher in ICB-sensitive or responder mice compared to ICB-resistant or non-responder mice (two-tailed Wilcoxon rank-sum test; Figure R1B, C; Supplementary Figure 7B, C). Although statistical significance was not reached

due to limited sample size, the trends were robust and aligned with our observations from peripheral blood.

Furthermore, tumors from ICB-sensitive mice showed significantly greater T cell clonal expansion following anti-PD-1 treatment compared to resistant tumors and both T and B cells displayed increased clonal expansion in responders relative to non-responders. (two-tailed Wilcoxon rank-sum test; Figure R1D-F; Supplementary Figure 7D-F), further supporting a link between intratumoral and peripheral immune activity. These results suggest that the blood-based LiBIO score reflects meaningful intratumoral immune dynamics during ICB treatment. This provides biological validation of the score and strengthens its translational relevance.

To reflect this addition, we have included the following paragraph in the revised Results section of the manuscript:

“To assess whether dynamic changes in blood reflect corresponding immune activity within the tumor microenvironment, we first generated in-house bulk RNA-seq data from tumors of HNSCC mice treated with anti-PD-1. Tumor biopsies were collected 14 days after cell implantation from five responders and four non-responders (**SF 7A**). We also analyzed an independent murine dataset in which single-cell RNA-seq and TCR-seq were performed on tumors from ICB-sensitive and ICB-resistant HNSCC models. Across both datasets, effector memory T cell, B cell, and LiBIO scores were elevated in ICB-sensitive or responder tumors compared to resistant or non-responder tumors (two-tailed Wilcoxon rank-sum test; **SF 7B,C**). While not statistically significant due to limited sample size, the trends were consistent and biologically meaningful. Notably, in the ICB-sensitive tumors, T cell clonal expansion significantly increased following treatment and both T and B cells displayed increased clonal expansion in responders relative to non-responders (two-tailed Wilcoxon rank-sum test; **SF 7D-F**), mirroring the dynamics observed in the peripheral blood. These results suggest that the clonal expansion detected in the blood is reflective of tumor-intrinsic immune activation and support the notion that blood-derived immune signatures, such as the LiBIO score, capture biologically meaningful tumor immune dynamics during ICB therapy.”

Figure R1 (Supplementary Figure 7 in revised manuscript). Distribution of immune signature scores and T cell clonal expansion in ICB-sensitive and ICB-resistant mice. **A.** Tumor volume trajectories in responder (blue lines) and non-responder (red lines) mice following anti-PD-1 treatment. **B.** Effector memory T cell (T_{em}), B cell, and LIBIO scores in responder versus non-responder mice based on in-house tumor RNA-seq data. **C.** T_{em} , B cell, and LiBIO scores in ICB-sensitive (MOC1) and ICB-resistant (MOCe1) mouse models from a publicly available dataset. **D–E.** Clonal expansion of T cells (**D**) and B cells (**E**) in responder and non-responder mice. **F.** T cell clonal expansion in ICB-sensitive mice (MOC1), represented as \log_2 -transformed clonal sizes before (green boxes) and after (orange boxes) anti-PD-1 treatment. Statistical significance was evaluated using the Wilcoxon rank-sum test and is indicated as follows: $P < 0.05$ (\ast), $P < 0.01$ ($\ast\ast$), $P < 0.001$ ($\ast\ast\ast$), $P \geq 0.05$ (ns).

2. The study focuses specifically on ICB response in HNSCC, which is clinically appropriate. However, it would be informative to examine whether the LiBIO score is applicable to other tumor types using publicly available datasets. A discussion on whether the immune landscape during ICB treatment in HNSCC is distinct from that in other cancers would help clarify whether the score has tumor-type specificity or broader utility.

Response:

We appreciate the reviewer's insightful comment. To assess the generalizability of the LiBIO score beyond HNSCC, we evaluated its predictive performance across 11 publicly available ICB-treated patient groups from six melanoma cohorts, three non-small cell lung cancer (NSCLC) cohorts, and two breast cancer cohorts. The LiBIO score demonstrated strong and consistent predictive performance across these datasets, achieving an average AUC of 0.80 ± 0.09 in melanoma, 0.73

± 0.23 in NSCLC, and 0.72 ± 0.10 in breast cancer cohorts (**Figure R2A**). These results support the broader applicability of the LiBIO score across diverse cancer types and suggest that the underlying immune mechanisms captured by the score are conserved across multiple tumor immune landscapes.

To reflect this important extension of our findings, we have we have added a dedicated section in the results and highlighted this point in the abstract:

In abstract:

"Furthermore, the LiBIO score generalizes to melanoma, non-small cell lung cancer, and breast cancer, demonstrating potential as a clinically applicable blood-based biomarker for ICB response across multiple cancer types."

In result section:

"T_{em} and B cells play critical roles in mediating ICB responses, not only in HNSCC, but also across multiple cancer types^{17,44-46}. To evaluate the broader applicability of the LiBIO score beyond HNSCC, we analyzed its predictive performance across 11 ICB-treated patient groups, including both pre-treatment and on-treatment samples, from four melanoma cohorts^{4,47-49}, three non-small cell lung cancer (NSCLC) cohorts⁵⁰⁻⁵², and two breast cancer cohorts^{53,54}. The LiBIO score achieved an average AUC of 0.80 ± 0.09 in melanoma, 0.73 ± 0.23 in NSCLC, and 0.72 ± 0.10 in breast cancer cohorts (**Figure 6A**). To further enhance clinical interpretability, we aimed to identify fixed decision thresholds for LiBIO score classification specific to each cancer type. For each data type, one cohort was designated as a training cohort to determine the optimal threshold that maximized the odds ratio for ICB response. The optimal thresholds identified were 0.196 for melanoma, 0.632 for NSCLC, and 0.383 for breast cancer (**SF 11A-C**). In the respective training cohorts, these thresholds achieved odds ratios of 4.7, 1.4, and 2.1 (**Figure 6B**). We then applied the same thresholds to the remaining cohorts of each cancer type to evaluate generalizability. In these testing cohorts, LiBIO maintained strong performance, achieving average odds ratios of 3.4 ± 0.34 in melanoma (excluding the Riaz et al. cohort due to RNA-later biopsy preparation, which differs from FFPE used in the other cohorts), 2.3 ± 0.14 in NSCLC, and 1.3 in breast cancer (**Figure 6B**). These results highlight the potential of the LiBIO score as a robust and clinically applicable biomarker for predicting ICB response across multiple tumor types."

Figure R2 (Figure 6 in revised manuscript). Prediction of immune checkpoint blockade (ICB) response using the LiBIO score across multiple cancer types.

A. Area under the curve (AUC) values for LiBIO score-based prediction of ICB response across 11 patient groups. The x-axis labels indicate both the dataset source and the timing of sample collection relative to anti-PD-1 therapy, where “Pre” denotes samples collected before treatment initiation and “Post” denotes samples collected after treatment had begun. **B.** Odds ratio (OR) for distinguishing responders from non-responders using fixed LiBIO thresholds specific to melanoma, NSCLC, and breast cancer (see **Methods**). Orange bars represent training cohorts used to determine the optimal threshold, and green bars correspond to independent validation cohorts.

3. The LiBIO score applies different thresholds for bulk RNA sequencing (0.201) and single cell RNA sequencing (0.424), which may complicate its use in clinical settings. It is important to clarify whether the lower threshold in bulk RNA data reflects technical noise or dilution of immune signals due to the presence of non-immune cells. Whether the expanding cohort will influence the threshold should also be considered. A percentile-based threshold may offer greater robustness and improve generalizability across patient populations.

Response:

We appreciate the reviewer’s thoughtful comment. To investigate the difference in LiBIO score thresholds, we first examined the distribution of LiBIO scores across both bulk and single-cell RNA-seq cohorts. As shown in Figure R3A (SF 11A in revised manuscript), single-cell cohorts tend to have higher LiBIO scores than bulk cohorts.

We then compared the estimated abundances of B and T cells in the two data types. In the single-cell cohorts, immune cell abundance was calculated as the proportion of B or T cells among total cells per sample. For the bulk RNA-seq cohorts, we estimated immune cell abundance using a deconvolution approach based on transcriptomic profiles (see **Methods**). As shown in Figure R3B (SF 11B in revised manuscript), both B and T cell abundances were significantly higher in single-cell datasets compared to bulk datasets. This discrepancy likely reflects technical differences between the two data types.

These differences help explain why the LiBIO threshold is higher for single-cell RNA-seq (0.424) than for bulk RNA-seq (0.201). Given these data-type-specific properties, we chose to apply fixed, absolute thresholds rather than percentile-based thresholds. While percentile-based thresholds can be useful for within-cohort comparisons, they require access to the full cohort distribution to interpret an individual patient's response, which limits clinical utility. In contrast, fixed thresholds are more readily applicable in a clinical setting where predictions must often be made on a per-patient basis.

We have added the following clarification to the revised manuscript:

In result section:

“To understand the differing thresholds between bulk and single-cell data, we analyzed the distribution of LiBIO scores across both data types. Single-cell cohorts exhibited higher LiBIO scores compared to bulk RNA-seq cohorts (**SF 11A**). This was accompanied by significantly higher estimated abundances of B and T cells in the single-cell cohorts (**SF 11B**) (**Methods**), likely due to technical differences. These observations help explain why the single-cell threshold (0.424) is higher than the bulk RNA-seq threshold (0.201).”

In method section:

“**Estimation of Immune Cell Abundance**

To estimate immune cell abundance across different data modalities, we applied complementary approaches tailored to single-cell and bulk RNA-seq datasets.

For single-cell RNA-seq cohorts, immune cell abundance was calculated directly based on cell type annotations. Specifically, we computed the relative abundance of B cells and T cells by dividing the number of annotated cells of each type by the total number of cells within each sample. Cell annotations were assigned using canonical markers and validated clustering from the original publications or our in-house pipeline.

For bulk RNA-seq cohorts, we estimated immune cell abundance using CODEFACS (COntident DEconvolution For All Cell Subsets)⁶⁷, a robust deconvolution framework that enables accurate inference of cell type-specific signals from bulk RNA-seq data. As the input reference signature, we used a curated single-cell RNA-seq dataset³⁶ from HNSCC, which includes well-characterized malignant, immune, and stromal cell populations. This reference was used to deconvolve bulk expression profiles and extract relative abundance estimates for B cells and T cells across samples. All deconvolution steps were performed using default parameters unless otherwise specified.”

Figure R3 (Supplementary Figure 11 in revised manuscript). Distribution of LiBIO scores and immune cell abundance across bulk and single-cell cohorts.

A. Density plot showing the distribution of LiBIO scores in ICB-treated patient cohorts profiled by bulk RNA-seq (blue) and single-cell RNA-seq (orange). Dashed vertical lines indicate the fixed thresholds used to define responders: 0.201 for bulk data and 0.424 for single-cell data. **B.** Comparison of B cell and T cell abundance between bulk and single-cell cohorts. The y-axis represents the estimated cell abundance, and each dot corresponds to a single sample. Cell abundances in single-cell cohorts were calculated based on the proportion of annotated cells per sample, while abundances in bulk cohorts were estimated using CODEFACS deconvolution with an HNSCC single-cell reference signature.

4. The predictive performance of the LiBIO score should be compared directly with established clinical biomarkers in HNSCC, such as the PD-L1 combined positive score (CPS). The absence of CPS in the benchmarking analysis reduces the translational impact of the study. Including CPS and exploring whether a combined biomarker strategy improves predictive accuracy would strengthen the clinical relevance of the findings.

Response:

We appreciate this thoughtful suggestion. Unfortunately, based on our current knowledge, we could not identify any publicly available ICB-treated HNSCC datasets that include both transcriptomic data and clinically measured PD-L1 combined positive scores (CPS). To address this gap, we utilized a publicly available single-cell RNA-seq dataset that contains both malignant and immune cells². This allowed us to estimate a CPS-like score based on the expression of CD274 (PD-L1) across tumor and immune compartments (as described in the Methods).

We then compared the predictive performance of the LiBIO score against this single-cell derived CPS score. As shown in **Figure R4**, the LiBIO score demonstrated superior predictive accuracy, as measured by ROC AUC. While this result supports the potential clinical value of LiBIO, we acknowledge that CPS here was approximated from transcriptomic data and not assessed via standard clinical immunohistochemistry. We have added this comparison and the associated

caveats to the revised manuscript, and we highlight the need for future validation in datasets that include clinically annotated CPS scores.

Results section addition:

“To benchmark the LiBIO score against FDA-approved biomarkers, we next compared it to the PD-L1 combined positive score (CPS). Due to the lack of publicly available ICB-treated HNSCC datasets with both transcriptomic profiles and clinical CPS annotations, we estimated CPS using a publicly available single-cell RNA-seq dataset (**Methods**) previously incorporated in this study³⁷. This dataset contains both malignant and immune cells, allowing for transcriptomic estimation of CPS based on CD274 expression. The LiBIO score showed superior predictive performance compared to the estimated CPS score in ROC analysis (**SF 8A, B**). Nonetheless, because CPS was inferred from transcriptomic data rather than immunohistochemistry, further validation in datasets with clinically measured CPS is warranted.”

Discussion section addition:

“The resulting LiBIO score integrates the T_{em} and B cell signatures, and quite remarkably, its application using one fixed decision threshold yields a fairly high predictive odds ratio of response, which surpasses the predictive power of existing biomarkers and a transcriptomic based CPS score. Although this finding is encouraging, we emphasize that the CPS score was derived from RNA expression data and not standard clinical IHC measurements.”

Method section addition:

“Calculation of PD-L1 combined positive score from single-cell RNA-seq data

To approximate the clinically defined PD-L1 Combined Positive Score (CPS) using single-cell RNA sequencing (scRNA-seq) data, we computed a transcriptomic analog based on the expression of *CD274*, which encodes PD-L1.

Cell annotations were obtained from the original publication and used to classify each cell as either a tumor cell or an immune cell. Non-relevant cell types (e.g., stromal or endothelial cells) were excluded from downstream analysis.

A cell was considered PD-L1 positive if more than two reads were mapped to the *CD274* gene. This threshold was chosen to minimize potential noise from low-level or spurious expression often observed in single-cell RNA-seq data. For each sample, the CPS was calculated using the following formula:

$$CPS = \frac{\text{Number of } PDL1^+ \text{ tumor cells} + \text{Number of } PDL1^+ \text{ immune cells}}{\text{Total number of tumor cells}} \times 100$$

”

Figure R4 (Supplementary Figure 8 in revised manuscript). Prediction of ICB response in HNSCC using the LiBIO and CPS scores.

A. Boxplots showing the distribution of LiBIO and estimated CPS scores (based on CD274 expression) between ICB responders and non-responders. Statistical comparisons were made using two-tailed Wilcoxon rank-sum tests. **B.** ROC curves comparing the predictive performance of the LiBIO score and transcriptomic based CPS score. Statistical significance is denoted as follows: $P < 0.05$ (*), $P < 0.01$ (**), $P < 0.001$ (***), $P \geq 0.05$ (ns).

Reviewer #2 (Remarks to the Author): expertise in HNSCC multi-omics and TCR-seq

In this interestingly titled study by Wang et al, the authors use a previously established syngeneic mouse model of HNSCC, treat these with anti-PD1 blockade and obtain blood from these mice at 4 different time points to determine: bulk RNAseq, scRNAse and scTCRseq from PBMCs. The main objective here is to establish the dynamics of PMBC comparing responders with non-responder mice. This is a nice system as it appears that a third of the mice appear to respond to the treatment while 2/3rds do not, but little detail is provided on the mouse system used. I had to trawl through this and previous papers to even understand the model, and I am uncertain given the single genetic background of a cell line and the mouse, how clonality varies between the responders and non-responders to account for this differential response. Furthermore, I am also concerned that the entire basis of this paper is on a single mouse line (albeit very comprehensively analysed), and I feel that further validation on a different system would be critical, especially since this data covers 4/5 of the figures in the paper; at the very least the critical parts of the study, time points and validation of the Tem, B and clonal dynamics that are presented here.

Response:

We thank the reviewer for raising this important point. To address the concern about model transparency and response heterogeneity, we now provide additional detail on the 4MOSC1 syngeneic orthotopic HNSCC model used in this study. This model, first described in Wang et

al., Nat Comm 2019³ and further expanded in Saddawi-Konefka et al., Nat Comm 2022⁴, is derived from tobacco carcinogen exposure and exhibits partial but reproducible sensitivity to α PD-1 monotherapy (response rate ~30–40%). Its orthotopic site, immune competence, and clinically relevant mixed-response phenotype have made it a widely used platform for immunotherapy studies in HNSCC.

The observed variation in tumor response within this single model is not driven by clonal heterogeneity in the tumor cell line, which is genetically stable, but rather reflects dynamic differences in immune priming, lymphatic function, and host-intrinsic response programs, as shown in prior work (Saddawi-Konefka et al., Nat Comm 2022). To further address the reviewer's concern, we have now analyzed tumors from responders and non-responders in our cohort using RNA-seq to evaluate clonal and transcriptional features. We found that T and B cells displayed increased clonal expansion in responders relative to non-responders.

Regarding the use of a single preclinical model, we fully agree that extending our findings to additional systems is important. To this end, we performed two sets of independent analyses to assess generalizability:

First, we evaluated whether our blood-derived signatures reflect immune activity within the tumor using an independent dataset of ICB-sensitive and ICB-resistant HNSCC mouse models (MOC1 vs. MOCe1). We observed that T_{em} , B cell, and LiBIO scores were elevated in tumors from ICB-sensitive mice, and T cell clonal expansion significantly increased following treatment, mirroring blood findings (**Figure R1**) (**Supplementary Figure 7 in revised manuscript**). These results reinforce the biological relevance of our blood-derived biomarker.

Second, to test broader applicability across tumor types, we analyzed 11 additional ICB-treated cohorts spanning melanoma, NSCLC, and breast cancer. The LiBIO score demonstrated strong predictive performance across all three cancer types (AUCs: melanoma 0.80 ± 0.09 , NSCLC 0.73 ± 0.23 , breast 0.72 ± 0.10), suggesting that the underlying immune mechanisms are conserved beyond HNSCC (**Figure R2**) (**Figure 6 in revised manuscript**).

These additions have been integrated into the manuscript as described below.

Manuscript Addition:

Add at the end of the Study Overview paragraph in the **Results** section:

"The 4MOSC1 model used in this study is a well-characterized, carcinogen-induced, orthotopic HNSCC model that exhibits a reproducible mixed-response to anti-PD-1 therapy (~30–40%), consistent with clinical response rates observed in human HNSCC. This model was originally described in Wang et al., Nat Comm 2019²⁷ and further refined in Saddawi-Konefka et al., Nat Comm 2022²⁶, where we demonstrated that response heterogeneity is driven by differences in immune activation and lymphatic function, rather than tumor-intrinsic clonal variation."

"To assess whether dynamic changes in blood reflect corresponding immune activity within the tumor microenvironment, we first generated in-house bulk RNA-seq data from tumors of HNSCC mice treated with anti-PD-1. Tumor biopsies were collected 14 days after cell implantation from five responders and four non-responders (SF 7A). We also analyzed an independent murine

dataset in which single-cell RNA-seq and TCR-seq were performed on tumors from ICB-sensitive and ICB-resistant HNSCC models. Across both datasets, effector memory T cell, B cell, and LiBIO scores were elevated in ICB-sensitive or responder tumors compared to resistant or non-responder tumors (two-tailed Wilcoxon rank-sum test; SF 7B,C). While not statistically significant due to limited sample size, the trends were consistent and biologically meaningful. Notably, in the ICB-sensitive tumors, T cell clonal expansion significantly increased following treatment and both T and B cells displayed increased clonal expansion in responders relative to non-responders (two-tailed Wilcoxon rank-sum test; SF D-F), mirroring the dynamics observed in the peripheral blood. These results suggest that the clonal expansion detected in the blood is reflective of tumor-intrinsic immune activation and support the notion that blood-derived immune signatures, such as the LiBIO score, capture biologically meaningful tumor immune dynamics during ICB therapy.”

Figure R1 (Supplementary Figure 7 in revised manuscript). Distribution of immune signature scores and T cell clonal expansion in ICB-sensitive and ICB-resistant mice. A. Tumor volume trajectories in responder (blue lines) and non-responder (red lines) mice following anti-PD-1 treatment. **B.** Effector memory T cell (T_{em}), B cell, and LiBIO scores in responder versus non-responder mice based on in-house tumor RNA-seq data. **C.** T_{em} , B cell, and LiBIO scores in ICB-sensitive (MOC1) and ICB-resistant (MOC1e1) mouse models from a

publicly available dataset. **D–E.** Clonal expansion of T cells (**D**) and B cells (**E**) in responder and non-responder mice. **F.** T cell clonal expansion in ICB-sensitive mice (MOC1), represented as \log_2 -transformed clonal sizes before (green boxes) and after (orange boxes) anti-PD-1 treatment. Statistical significance was evaluated using the Wilcoxon rank-sum test and is indicated as follows: $P < 0.05$ (\ast), $P < 0.01$ ($\ast\ast$), $P < 0.001$ ($\ast\ast\ast$), $P \geq 0.05$ (ns).

" T_{em} and B cells play critical roles in mediating ICB responses, not only in HNSCC, but also across multiple cancer types^{17,44-46}. To evaluate the broader applicability of the LiBIO score beyond HNSCC, we analyzed its predictive performance across 11 ICB-treated patient groups, including both pre-treatment and on-treatment samples, from four melanoma cohorts^{4,47-49}, three non-small cell lung cancer (NSCLC) cohorts⁵⁰⁻⁵², and two breast cancer cohorts^{53,54}. The LiBIO score achieved an average AUC of 0.80 ± 0.09 in melanoma, 0.73 ± 0.23 in NSCLC, and 0.72 ± 0.10 in breast cancer cohorts (**Figure 6A**). To further enhance clinical interpretability, we aimed to identify fixed decision thresholds for LiBIO score classification specific to each cancer type. For each data type, one cohort was designated as a training cohort to determine the optimal threshold that maximized the odds ratio for ICB response. The optimal thresholds identified were 0.196 for melanoma, 0.632 for NSCLC, and 0.383 for breast cancer (**SF 11A-C**). In the respective training cohorts, these thresholds achieved odds ratios of 4.7, 1.4, and 2.1 (**Figure 6B**). We then applied the same thresholds to the remaining cohorts of each cancer type to evaluate generalizability. In these testing cohorts, LiBIO maintained strong performance, achieving average odds ratios of 3.4 ± 0.34 in melanoma (excluding the Riaz et al. cohort due to RNA-later biopsy preparation, which differs from FFPE used in the other cohorts), 2.3 ± 0.14 in NSCLC, and 1.3 in breast cancer (**Figure 6B**). These results highlight the potential of the LiBIO score as a robust and clinically applicable biomarker for predicting ICB response across multiple tumor types."

Figure R2 (Figure 6 in revised manuscript). Prediction of immune checkpoint blockade (ICB) response using the LiBIO score across multiple cancer types.

A. Area under the curve (AUC) values for LiBIO score-based prediction of ICB response across 11 patient groups. The x-axis labels indicate both the dataset source and the timing of sample collection relative to anti-PD-1 therapy, where "Pre" denotes samples collected before treatment

initiation and “Post” denotes samples collected after treatment had begun. **B. Odds ratio (OR) for distinguishing responders from non-responders using fixed LIBIO thresholds specific to melanoma, NSCLC, and breast cancer (see **Methods**).** Orange bars represent training cohorts used to determine the optimal threshold, and green bars correspond to independent validation cohorts.

The other concern I have with this model is that a 'clinical' response is apparent after 10 days, which is of course sooner than what I would expect for a human patient (which makes this model attractive), BUT poses a problem when trying to compare the dynamics here to human timelines. Furthermore, any biomarker AFTER clinical response is pretty useless as well, so data from timepoints 4 is AFTER a clinical response and hence not particularly relevant. Timepoint 2 is therefore the more interesting timepoints to focus on here. would be great if all the plots in figures 1 and 2 could be labeled by actual days of treatment rather Than time points 1,2,3,4, which is harder to follow.

Response:

We thank the reviewer for these thoughtful comments. We fully agree that the kinetics of clinical response in the mouse model differ from what would be expected in human HNSCC patients. Unfortunately, analogous matched time series data from tumors or blood of HNSCC patients are currently not available. We have now explicitly discussed this limitation in the revised manuscript:

“As with any study, our work has a few limitations. First and foremost, analogous time series data from tumors or blood of human HNSCC patients does not yet exist, which restricted our ability to directly compare the temporal dynamics observed in our mouse findings to human data.”

“We identified five days after ICB treatment as the early on treatment time point in mouse that is optimal for assessing ICB response in mouse. This strategy may also be applicable for HNSCC patients but obviously the best timing of such early on-treatment time point in humans requires further careful investigation.”

Regarding the concern about timepoint 4, we agree that evaluating biomarkers after a clinical response is less clinically relevant. Our inclusion of timepoint 4 aimed to provide a more complete picture of the dynamic changes occurring during the course of ICB treatment. However, we focused our major analyses and conclusions on earlier time points, particularly timepoint 2, where early predictive biomarkers are more meaningful.

In response to the reviewer’s helpful suggestion, we have revised all relevant figures label the x-axes with the actual days of treatment, rather than timepoints 1–4, to improve clarity and make the timelines easier to follow.

Other critical points to be addressed are as follows:

1. scRNAseq supports the role of Tem dynamics in overall response and enhanced in the

responder, but the role of B-cell activation and the dynamics presented are statistically weak and is a bit of an over claim here.

Response:

We appreciate the reviewer's important feedback. Our single-cell data indeed reveal that two immune cell populations, effector memory CD8+ T cells (T_{em}) and B cells, are associated with ICB response. However, the dynamics of these two populations differ notably.

T_{em} cell expansion primarily occurs in responders and reaches the largest difference between responders and non-responders at the late on-treatment time point (**Figure R3A, B**) (**Figure 2C, D in revised manuscript**). In contrast, B cell expansion occurs earlier, predominantly at the early on-treatment time point in responders, but is absent in non-responders, resulting in the greatest divergence between groups at this earlier post-treatment stage (**Figure R3C, D**) (**Figure 2E, F in revised manuscript**). These findings align with a previous study⁵ that demonstrated a predictive role for B cells in response to ICB in HNSCC.

We recognize the reviewer's concern regarding the statistical strength of the B cell findings. While the B cell expansion trend is evident, we have revised the manuscript to more cautiously present the conclusions regarding B cells, emphasizing the temporal difference and the need for further validation.

In summary, both T_{em} and B cells exhibit dynamic changes associated with response to ICB in HNSCC, albeit with distinct timing and magnitudes, and we have adjusted the text to reflect a more balanced interpretation.

Relevant revised text in the Results section:

"B cell abundance increased in both responders and non-responders following ICB treatment. There was no significant difference between responders and non-responders at the pre-treatment (time point 1) and late on-treatment time points (time points 4) (Fig. 2E). However, we observed a modest but statistically significant earlier increase in B cells among responders at the early on-treatment time point (time point 2) (one-tailed Wilcoxon rank-sum test, $P = 0.027$). In contrast, non-responders exhibited a delayed B cell accumulation that became apparent only at the middle on-treatment time point (Fig. 2F). These findings align with a previous study that demonstrated a predictive role for B cells in response to ICB in HNSCC²⁸."

Figure R3. Temporal changes in cell abundance in blood following ICB treatment. **A.** Abundance changes of effector memory CD8+ T cells in responders and non-responders across four time points. In the box plot, each dot represents one sample. Statistical significance was determined using a one-tailed Wilcoxon rank-sum test. The Mann-Kendall test was applied to assess whether abundance changed monotonically across time points. **B.** Abundance differences of effector memory CD8+ T cells between responders and non-responders. Statistical significance between the two groups was calculated using a one-tailed Wilcoxon rank-sum test. **C.** Abundance changes of B cells in responders and non-responders across four time points. In the box plot, each dot represents one sample. Statistical significance was determined using a one-tailed rank-sum test. The Mann-Kendall test was applied to assess whether abundance changed monotonically across time points. **D.** Abundance differences of B cells between responders and non-responders. Statistical significance between the two groups was calculated using a one-tailed Wilcoxon rank-sum test. For all panels, the X-axis represents time points, and the Y-axis represents the fraction of cells out of the total measured. *Statistical significance is denoted as follows: $P < 0.05$ (*), $P < 0.01$ (**), $P < 0.001$ (***), $P \geq 0.05$ (ns).*

2. Throughout the paper, the authors have chosen statistically weaker tests (like one tailed tests etc), and I wonder if this is because more robust test would weaken their 'p values'

Response:

We appreciate the reviewer's concern regarding statistical testing. In response, we have revised our manuscript to use two-tailed tests for several comparisons, including all benchmarking analyses comparing the predictive performance of the LiBIO score to existing biomarkers and transcriptomic signatures. This change ensures a more conservative and rigorous statistical interpretation. We note that the results reported remain statistically significant in 8 out of 12 comparisons between LiBIO and other biomarkers (**Figure R4A**) (**Fig. 5C in the revised manuscript**).

At the same time, we retained one-tailed tests for comparisons related to immune cell abundance and clonal expansion between responders and non-responders. This decision is based on prior biological knowledge and well-established hypotheses: namely, that ICB treatment is expected to induce expansion of T and B cells, particularly in responders. Under this directional hypothesis, a one-tailed test is statistically justified to assess whether responders show increased immune activity.

We have revised the manuscript accordingly.

"The AUC quantifying ICB prediction performance of combined score to the previously established transcriptomics-based signatures HNSCC patients. Two-tailed p-values were displayed at the top of each box and were calculated using Wilcoxon rank test to compare the control (LiBIO score) group with other signatures."

Figure R4. LiBIO predict ICB response in human patients.

A. The AUC quantifying ICB prediction performance of combined score to the previously established transcriptomics-based signatures HNSCC patients. Two-tailed p-values were displayed at the top of each box and were calculated using Wilcoxon rank test to compare the control (combined score) group with other signatures.

3. the described methodology for scRNAseq and scTCRseq is completely wrong- how could they align their mouse data to human hg38 reference genome for RNAseq or TCRseq?

Response:

Thank you for pointing out this regrettable error, for which we sincerely apologize. A correction had made in the manuscript:

“Single-cell TCR sequencing data were aligned to the mouse mm10 V(D)J reference genome using Cell Ranger to obtain clonotype information for each cell.”

4. for the TCRseq, the results seem to favour BCRseq as a biomarker, especially in bulk sequencing...would be great if the authors expanded on this or explained it

Response:

We thank the reviewer for this insightful comment. We agree that our results indicate that B cell clonal expansion provides superior predictive power compared to T cell clonal expansion. We have expanded our explanation in the revised manuscript, as detailed below.

Our single-cell data demonstrate that the most significant difference in B cell abundance between responders and non-responders occurs at the early on-treatment time point (**Figure R3D**) (**Figure 2F in the manuscript**). Consistently, analysis of bulk RNA-seq data revealed that B cell expansion is strongest at early on-treatment time point (Day 9) compared to later on-treatment timepoints (**Figure R5A**) (**Figure 3F in the revised manuscript**). Furthermore, the area under the curve (AUC) values for predicting ICB response, based on clonal expansion scores, reveal that B cell clonal expansion is a better indicator of ICB efficacy than T cell clonal expansion at early on-treatment time points (**Figure R5B**) (**Figure 3G in the revised manuscript**). These findings support the notion that early treatment-induced B cell changes are most predictive of ICB response. We have clarified this observation in the revised manuscript and highlighted its association with ICB response.

As described in the revised text:

“However, we observed a modest but statistically significant earlier increase in B cells among responders at the early on-treatment time point (Day 9) (one-tailed Wilcoxon rank-sum test, $P = 0.027$). In contrast, non-responders exhibited a delayed B cell accumulation that became apparent only at the middle on-treatment time point (**Fig. 2F**). These findings align with a previous study²⁸ that demonstrated a predictive role for B cells in response to ICB in HNSCC.

“BCR analysis using bulk RNA sequencing data further demonstrated significant B cell clonal expansion following ICB treatment (**Fig. 3F**)”

“Interestingly, both T and B cell clonal expansion at early on-treatment time points exhibit strong predictive power for ICB response (**Fig. 3G**). Specifically, B cell clonal expansion demonstrates superior predictive ability compared to T cell clonal expansion at early on-treatment time points (**Fig. 3G**). This aligns with the timing of tumor shrinkage initiation in the responders (**Fig. 1C, D**) as well as the changes in cell abundance estimated from the single-cell data (**Fig. 2D, F**). Notably, B cell clonal expansion is predictive not only at the pre-treatment time point but also at two on-treatment time points, further underscoring the critical role of B cells in mediating the ICB response in HNSCC (**Fig. 3G**) (**SF 4 A-D**).”

Figure R5. ICB treatment induces T cell and B cell clonal expansion.

A. Comparison of B cell clonal expansion estimated using bulk RNA-seq data, between responders and non-responders. The y-axis represents the scaled Simpson index, where higher values indicate greater clonal expansion. Each dot represents an individual sample. Statistical significance was determined using a one-tailed Wilcoxon rank-sum test. **B.** AUC (Area Under the Curve) values for predicting ICB response based on the bulk B cell clonal expansion index, bulk T cell clonal expansion index, and single-cell T_{em} cell clonal expansion index at Day 9. *Statistical significance is denoted as follows: $P < 0.05$ (*), $P < 0.01$ (**), $P < 0.001$ (***), $P \geq 0.05$ (ns).*

5. As stated above, showing a good AUC after seeing a clinical response (timeline 4) is not useful, and neither is the dynamics between T4-T2. I would focus more on T2 as this pre-dates any clinical response.

Response:

We thank the reviewer for this important point. We fully agree that biomarkers intended to predict ICB response must pre-date any observable clinical response to be of true clinical utility. Accordingly, we agree that the early on-treatment time point (timepoint 2, Day 9) is the most critical and relevant window for prediction.

We have initially included analyses of timepoint 4 and T4–T2 dynamics to provide a comprehensive view of immune changes over time, but we of course recognize that their clinical relevance is limited because tumor shrinkage is already apparent by these later stages. Following your advice we have now revised the manuscript to place greater emphasis on timepoint 2 in both the relevant sections, and we have clarified that timepoint 2 pre-dates any measurable clinical response.

Specifically, we now state in the manuscript:

"The early on-treatment time point (Day 9) emerges as the optimal window for assessing treatment efficacy, as it precedes observable tumor shrinkage and captures critical early immune dynamics associated with response."

6. Validation was done on data and timepoints that were available, and likely could have missed all the early timelines here. Also I would have liked to have seen PBMC TCR and BCR dynamics validation. Do the numbers really go up and prune off as quickly as suggested here?

Response:

We thank the reviewer for this important point. Unfortunately, we could not identify publicly available patient datasets that include PBMC TCR or BCR profiling with sufficient temporal resolution during early ICB treatment, which limits direct validation of T and B cell dynamics in the peripheral blood of human patients.

To address this question indirectly, we analyzed a publicly available mouse HNSCC model dataset¹. In this study, the authors used the ICB-sensitive MOC1 tumor model and performed single-cell RNA-seq and TCR-seq analysis of tumor-infiltrating T cells before and after antiPD1 therapy. Notably, tumor shrinkage in responder mice began around day 10 post-inoculation (their Figure 1B), which aligns closely with our day 9 sampling timepoint (timepoint 2) where we observed B and T cell expansion in the blood.

Using the TCR data from this model, we observed that T cell clonal expansion was significantly increased after antiPD1 treatment in responders (**Figure R1C**) (**Supplementary Figure 7C in revised manuscript**), providing supporting evidence that early T cell expansion in response to ICB is biologically plausible and occurs within a comparable timeframe.

This analysis has been added to the revised manuscript.

"Notably, in the ICB-sensitive tumors, T cell clonal expansion significantly increased following treatment (two-tailed Wilcoxon rank-sum test; **SF 7C**), mirroring the dynamics observed in the peripheral blood."

7. Trying to validate the nice blood based biomarker on tumor datasets and TCGA (to correlate with OS), is a bit regressive. The validation should be like for like. The latter data just suggests to me that you have an all-encompassing immune marker that is overall prognostic rather than predictive. If there is no difference to other scoring system, it should be stated as such. Similarly,

I do. not understand the continual used of a single-tail statistical test. Sounds like you have already decided your model is better!

Response:

We appreciate the reviewer's comments and the opportunity to clarify our rationale. Our primary focus was to develop and validate the LiBIO score as a blood-based predictive biomarker for ICB response, and we initially demonstrated its performance using multiple peripheral blood cohorts. Due to the limited availability of large, well-annotated blood-based datasets, and considering the well-established biological premise that T and B cells can traffic between tumors and peripheral blood, we extended our evaluation of LiBIO to tumor-derived transcriptomic datasets, including both ICB-treated and ICB-naïve cohorts.

Our hypothesis was that if LiBIO captures key immune dynamics, it may also retain predictive or prognostic value when applied to tumor data. In our benchmarking analysis using tumor-based HNSCC cohorts, LiBIO outperformed or performed comparably to other state-of-the-art transcriptomic signatures in predicting ICB response (**Figure R4A**) (**Figure 5C in revised manuscript**). We have now clarified in the revised manuscript that this finding does not suggest LiBIO is universally superior, but rather that it has robust performance across both blood and tumor data types.

"Subsequently, we assessed the predictive efficacy of the LiBIO score across patient cohorts, comparing its performance to several contemporary transcriptomics-based biomarkers^{15,16,18,19,38-43}. Overall, the LiBIO score exhibited superior predictive performance relative to alternative biomarkers (**Fig. 5C**). Interestingly, two functional CD8+ T cell related tumor signatures also performed strongly in bulk RNA sequencing cohorts, highlighting the pivotal role of functional CD8+ T cells in mediating ICB responses. However, while these functional CD8+ T cell related signatures performed well in bulk datasets, their performance was less consistent in single-cell cohorts. In contrast, the LiBIO score demonstrated greater robustness and predictive accuracy across both bulk and single-cell RNA sequencing datasets (**Fig. 5C**)."

We further evaluated the prognostic potential of LiBIO in both ICB-treated and ICB-naïve datasets (e.g., TCGA), to distinguish between predictive and prognostic roles. The TCGA analysis was specifically included to assess whether LiBIO is associated with general immune activation and survival, even in the absence of ICB treatment (**Figure R6D**) (**Figure 5D in revised manuscript**). We now explicitly clarify this distinction and its limitations in the revised discussion.

"Additionally, survival analyses revealed that patients with higher LiBIO scores had significantly improved survival outcomes compared to those with lower scores. This predictive association was more pronounced in ICB-treated cohorts and remained significant in ICB-treatment-naïve datasets (**Fig. 5D**) (**SF 9A-C**). Importantly, although the signatures were derived from an HPV-negative HNSCC mouse model, they remained significantly correlated with patient survival after adjusting for HPV status (**Fig. 5D**)."

“The LiBIO score not only demonstrates predictive utility for ICB response in both blood and tumor transcriptomic datasets but also exhibits prognostic value in both ICB-treated and ICB-naïve cohorts. While this suggests that LiBIO reflects core features of anti-tumor immune activation, future studies are needed to disentangle its predictive versus prognostic roles in distinct clinical contexts.”

Regarding the use of one-tailed statistical tests, we reiterate (as addressed in Comment 2) that we used one-tailed tests only in biologically directional hypotheses (e.g., immune expansion in responders), while all performance benchmarking and survival analyses were evaluated using two-tailed tests. We have clarified this consistently throughout the manuscript.

“The AUC quantifying ICB prediction performance of combined score to the previously established transcriptomics-based signatures HNSCC patients. Two-tailed p-values were displayed at the top of each box and were calculated using Wilcoxon rank test to compare the control (LiBIO score) group with other signatures.”

8. The discussion should focus on comparing these markers to other established markers of PD1 response, some of which the senior author has co-written and shows much better AUC than this study.

Response:

We appreciate the reviewer’s important suggestion. We first compared the LiBIO score with the PD-L1 combined positive score (CPS), the FDA-approved biomarker for predicting ICB response in HNSCC. Due to the absence of HNSCC datasets with both transcriptomic data and clinically assessed CPS, we estimated CPS from a publicly available single-cell RNA-seq dataset previously used in our study¹. This allowed us to evaluate CD274 expression across tumor and immune cells. We then compared the predictive performance of the LiBIO score against this single-cell derived CPS score. As shown in **Figure R6 (SF 8A, B in revised manuscript)**, the LiBIO score demonstrated superior predictive accuracy, as measured by ROC AUC. While this result supports the potential clinical value of LiBIO, we acknowledge that CPS here was approximated from transcriptomic data and not assessed via standard clinical immunohistochemistry. We have added this comparison and the associated caveats to the revised manuscript, and we highlight the need for future validation in datasets that include clinically annotated CPS scores.

Results section addition:

“To benchmark the LiBIO score against FDA-approved biomarkers, we next compared it to the PD-L1 combined positive score (CPS). Due to the lack of publicly available ICB-treated HNSCC datasets with both transcriptomic profiles and clinical CPS annotations, we estimated CPS using a publicly available single-cell RNA-seq dataset (**Methods**) previously incorporated in this study³⁷. This dataset contains both malignant and immune cells, allowing for transcriptomic estimation of CPS based on CD274 expression. The LiBIO score showed superior predictive performance compared to the estimated CPS score in ROC analysis (**SF 8A, B**). Nonetheless,

because CPS was inferred from transcriptomic data rather than immunohistochemistry, further validation in datasets with clinically measured CPS is warranted.”

Discussion section addition:

“The resulting LiBIO score integrates the T_{em} and B cell signatures, and quite remarkably, its application using one fixed decision threshold yields a fairly high predictive odds ratio of response, which surpasses the predictive power of existing biomarkers and a transcriptomic based CPS score. Although this finding is encouraging, we emphasize that the CPS score was derived from RNA expression data and not standard clinical IHC measurements.”

Figure R6. Prediction of ICB response in HNSCC using the LiBIO and CPS scores.

A. Boxplots showing the distribution of LiBIO and estimated CPS scores (based on CD274 expression) between ICB responders and non-responders. Statistical comparisons were made using two-tailed Wilcoxon rank-sum tests. **B.** ROC curves comparing the predictive performance of the LiBIO score and transcriptomic based CPS score. Statistical significance is denoted as follows: $P < 0.05$ (*), $P < 0.01$ (**), $P < 0.001$ (***), $P \geq 0.05$ (ns).

We then extended our analysis to include two additional widely used transcriptome-based biomarkers, TIDE and IMPRES, bringing the total number of biomarkers evaluated to twelve. We benchmarked LiBIO against these biomarkers across multiple HNSCC datasets and found that LiBIO performs comparably or favorably, depending on the cohort (**Figure R7A**) (**Figure 5C in the revised manuscript**).

We have revised the manuscript accordingly.

“Overall, the LiBIO score exhibited superior predictive performance relative to alternative biomarkers (**Fig. 5C**). Interestingly, two functional CD8+ T cell related tumor signatures also performed strongly in bulk RNA sequencing cohorts, highlighting the pivotal role of functional CD8+ T cells in mediating ICB responses. However, while these functional CD8+ T cell related

signatures performed well in bulk datasets, their performance was less consistent in single-cell cohorts. In contrast, the LiBIO score demonstrated greater robustness and predictive accuracy across both bulk and single-cell RNA sequencing datasets (Fig. 5C). ”

Figure R5. Prediction of ICB response in HNSCC using the LiBIO and CPS scores.

A. The AUC quantifying ICB prediction performance of combined score to the previously established transcriptomics-based signatures HNSCC patients. Two-tailed p-values were displayed at the top of each box and were calculated using Wilcoxon rank test to compare the control (combined score) group with other signatures.

Reviewer #3 (Remarks to the Author): expertise in HNSCC mouse models

The paper aims to look at biomarkers in a mouse model of HNSCC to determine if there is a pattern of changes in the blood that can determine whether a patient will elicit a positive response to the administration of immune checkpoint blockade (ICB). The authors utilize single cell and bulk mRNA signatures.

The authors find differences in clonal expansion and temporal dynamics of effector T cells and B cells between responder and non-responder mice. These changes were seen early in treatment. A LiBIO score (Liquid Biomarker of Immunotherapy Outcome) from these data predicts ICB response in HNSCC patients. The authors propose these methods as a non-invasive, biologically grounded approach that can predict ICB outcomes with high fidelity. They suggest this method can be used to determine biomarkers for other cancer types.

The authors use a 4MOSC cell line, a carcinogen-induced orthotopic tumor HNSCC model where tumors in c57bl6 mice were induced by 4-nitroquinoline-1 oxide (4NQO), implanted into

the tongue (1×10^5 cells). Blood was drawn at days 4, 9, 17- and 24-days post implant. PD-1 treatment was started immediately after the 4 day collection. Treatment efficacy was determined by tumor growth and immune responses. Bulk RNA-seq, scRNA-seq and scTCR sequencing was utilized to examine immune repertoire changes.

The authors created 2 data sets of 45, 16 mice cohorts with the sets split into responders and non-responders. They also incorporated 20 HNSCC patient samples from ICB trials. They also accessed two previously published ICB treatment data sets and 4 publicly available ICB bulk RNA-seq data sets.

The in-vivo murine data showed an increase in CD4, Cd8 and B cell abundance after treatment compared to pre-ICB and a loss of neutrophils. When examining subsets of the increased cells effector memory cells Tem and Th1 cells were increased in each time point after ICB compared to pre-treatment which was not seen in non-responders. A kinetic difference in the accumulation of B-cells was also observed.

Clonal expansion of T and B cells were increased in responders after ICB compared to non-responders. Tem cells has a population that increased post ICB that was not seen in non-responders. This was validated utilizing a different cohort and using bulk RNAseq instead of scRNAseq showing T-cell expansion occurred early.

These early effects were predictive of response, with Bcells showing predictive efficacy at the 3 earliest time points (1,2,3).

The changes in biological functions were assessed by algorithmic determination of 8 clusters of genes that are differentially regulated in the responders (5 upregulated compared to pretreatment including B-cell activation, immunoglobulin production and mitochondrial gene expression) and downregulated in the non-responders (6 down regulated including bcell activation and immunoglobulin production). Non-responders showed downregulation in innate immune response, leukocyte-mediated cytotoxicity, and cell killing were downregulated.

The authors generated a gene signature for CD8+ t cells (184 genes) and B cells (168 genes) and utilized the signatures for their predictive measure testing them against 7 ICB treated cohorts. The authors find significance in predicting the ICB response with AUC values of 0.96 and 0.74 for T cell and B cell clusters, respectively. They combine the signatures into a LiBIO score that improved the AUC value to 0.8.

These predictive tests were performed on human databases finding AUC values similar to the mouse data of 0.75, 0.74 and 0.78 for the T cell, B cell and combined signatures. This was summarized to be an improvement on other biomarkers.

Comments

It's an in-depth study of the effect of ICB on the immune abundance in patient blood before and after treatment. The findings are exciting from a diagnostic angle but also eludes to the

differences in patients that respond to ICB highlighting the importance of T cell effector cells and B cells in HNSCC.

The study appears to be run in a well-designed fashion with the authors utilizing several different but parallel cohorts to determine the role of the cell signatures.

The authors acknowledge the limitations of the study; lack of analogous time series data from human patients, very few studies with PBMC data and small sample sizes.

The paper is well written with very few errors in grammar or spelling.

Reviewer #3 (Remarks on code availability):

Comments

It's an in-depth study of the effect of ICB on the immune abundance in patient blood before and after treatment. The findings are exciting from a diagnostic angle but also eludes to the differences in patients that respond to ICB highlighting the importance of T cell effector cells and B cells in HNSCC.

The study appears to be run in a well-designed fashion with the authors utilizing several different but parallel cohorts to determine the role of the cell signatures.

The authors acknowledge the limitations of the study; lack of analogous time series data from human patients, very few studies with PBMC data and small sample sizes.

The paper is well written with very few errors in grammar or spelling.

Response:

We are grateful for your recognition of the study's design, the translational potential of our findings, and the relevance of the LiBIO score for ICB response prediction in HNSCC.

For your information, we have implemented several major revisions in response to the detailed comments from the other reviewers. These include:

- Benchmarking LiBIO against a few established ICB response biomarkers such as CPS, TIDE, and IMPRES,
- Validating LiBIO's performance in tumor tissue using a publicly available ICB-sensitive HNSCC mouse model,
- Evaluating LiBIO's predictive performance in additional cancer types and treatment timepoints,
- Further clarifying the distinction between predictive and prognostic roles of the score,

- And refining the statistical methodology throughout.

We believe these revisions have substantially improved the clarity, rigor, and impact of our study. Thank you again for your positive feedback and support of our work, which is much appreciated.

Reference

1. Zhou, L. *et al.* Checkpoint blockade-induced CD8+ T cell differentiation in head and neck cancer responders. *J. Immunother. Cancer* **10**, e004034 (2022).
2. Bill, R. *et al.* CXCL9:SPP1 macrophage polarity identifies a network of cellular programs that control human cancers. *Science* **381**, 515–524 (2023).
3. Wang, Z. *et al.* Syngeneic animal models of tobacco-associated oral cancer reveal the activity of in situ anti-CTLA-4. *Nat. Commun.* **10**, 5546 (2019).
4. Saddawi-Konefka, R. *et al.* Lymphatic-preserving treatment sequencing with immune checkpoint inhibition unleashes cDC1-dependent antitumor immunity in HNSCC. *Nat. Commun.* **13**, 4298 (2022).
5. Chang, T.-G. *et al.* Tumor and blood B-cell abundance outperforms established immune checkpoint blockade response prediction signatures in head and neck cancer. *Ann. Oncol.* S0923753424049147 (2024) doi:10.1016/j.annonc.2024.11.008.

Responses to reviewer comments

Within this document, the reviewer comments are copied verbatim in black font and alternate with our responses, which are presented in red font. Any text additions made to the manuscript to address the reviewers' comments are highlighted in turquoise and are indented, except for very short additions.

Reviewer #1 (Remarks to the Author):

The revised manuscript adequately addresses the questions raised by the reviewers with strong additional data. The current version is more solid and concrete. I have no further comments on the manuscript, and I consider it acceptable for publication.

Response:

We thank the reviewer for the positive feedback and support for publication.

Reviewer #2 (Remarks to the Author):

Thank you for the revised manuscript, addressing my comments and newer data added. I appreciate the addition of a second mouse model to validate the data and additional human data beyond HNSCC, where the Libio scoring system seems to hold true as well. There are some minor errors in labelling, colour and legends that should be corrected as well prior to publication (ugh app figure 7a, where the legend says red and blue lines, while the figure is blue and yellow)..these are minor but it is good to get these right, because a lot of data is in the supplementary section which is useful as a resource for other studies. Apart from that I support the publication of this manuscript. Congratulations to all the investigators...

Response:

We thank the reviewer for the positive feedback and support for publication. The legend for Supplementary Figure 7A has been revised. Additionally, data related to the main figures and supplementary figures have been provided as Source Data with the manuscript.

“Supplementary Figure 7. Distribution of immune signature scores and T cell clonal expansion in ICB-sensitive and ICB-resistant mice.

A. Tumor volume trajectories in responder (blue lines) and non-responder (yellow lines) mice following anti-PD-1 treatment.”

Reviewer #3 (Remarks to the Author):

Thank you for your response to our comments.

Response:

We thank the reviewer for the positive feedback and support for publication.

Reviewer #3 (Remarks on code availability):

I do not know how to interpret any of this code

Response:

The Code Availability section has been revised accordingly. The code has been deposited in GitHub and archived in Zenodo to ensure reproducibility and citation. Both the GitHub link and Zenodo DOI have been included in the manuscript.

“Code availability

All original code used in this study has been deposited in GitHub at https://github.com/wbb1813/Time_series_mouse_ICB and is publicly available as of the date of publication. To ensure reproducibility and provide a permanent citation, the repository has also been archived in Zenodo with the <https://doi.org/10.5281/zenodo.15856815>.”